**Hydro-ecological controls on dissolved carbon dynamics in groundwater and export to streams in a temperate pine forest**

Loris Deirmendjian[1], Denis Loustau[2], Laurent Augusto[2], Sébastien Lafont[2], Christophe Chipeaux[2], Dominique Poirier[1], and Gwenaël Abril[1, 3, *].

[1]Laboratoire Environnements et Paléoenvironnements Océaniques et Continentaux (EPOC), CNRS, Université de Bordeaux, Allée Geoffroy Saint-Hilaire, 33615 Pessac Cedex France.

[2]INRA, UMR 1391 Interactions Sol-Plante-Atmosphère (ISPA), 33140 Villenave d'Ornon, France.

[3]Departamento de Geoquímica, Universidade Federal Fluminense, Outeiro São João Batista s/n, 24020015, Niterói, RJ, Brazil.

[*]Now also at Laboratoire d'Océanographie et du Climat, Expérimentations et Approches Numériques (LOCEAN), Centre IRD France-Nord, 32, Avenue Henri Varagnat, F-93143 Bondy, France.

Correspondence to Loris Deirmendjian: lorisdeir@gmail.com

**Abstract**. We studied the export of dissolved inorganic carbon (DIC) and dissolved organic carbon (DOC) from forested shallow groundwater to first-order streams, based on groundwater and surface water sampling and hydrological data. The selected watershed was particularly convenient for such study, with a very low slope, with pine forest growing on sandy permeable podzol and with hydrology occurring exclusively through drainage of shallow groundwater (no surface runoff). A forest plot was instrumented for continuous eddy covariance measurements of precipitation, evapotranspiration and net ecosystem exchanges of sensible and latent heat fluxes as well as $CO_2$ fluxes. Shallow groundwater was sampled in 3 piezometers located in different plots and surface waters were sampled in 6 first-order streams; river discharge and drainage were modeled based on 4 gauging stations. On a monthly basis and at the plot scale, we found a good consistency between precipitation on the one hand and the sum of evapotranspiration, shallow groundwater storage and drainage on the other hand. DOC and DIC stocks in groundwater and exports to first-order streams varied drastically during the hydrological cycle, in

relation with water table depth and amplitude. In the groundwater, DOC concentrations were maximal in winter when the water table reached the superficial organic-rich layer of the soil. In contrast, DIC (in majority excess $CO_2$) in groundwater, showed maximum concentrations at low water table during late summer, concomitantly with heterotrophic conditions of the forest plot. Our data also suggests that a large part of the DOC mobilized at high water table was mineralized to DIC during the following months within the groundwater itself. In first-order streams, DOC and DIC followed an opposed seasonal trend similar to groundwater but with lower concentrations. On an annual basis, leaching of carbon to streams occurred as DIC and DOC in similar proportion, but DOC export occurred in majority during short periods of highest water table, whereas DIC export was more constant throughout the year. Leaching of forest carbon to first-order streams represented a small portion (approximately 2%) of the net land $CO_2$ sink at the plot. In addition, approximately 75% of the DIC exported from groundwater was not found in streams, as it returned very fast to the atmosphere through $CO_2$ degassing.

I. **Introduction**

Since the beginning of the Industrial Era, human activities have greatly modified the fluxes of carbon between the atmosphere and the continents, as well as those occurring along the aquatic continuum that connect the land and the coastal ocean (Cole et al., 2007; Ciais et al., 2013; Regnier et al., 2013). Globally, the land (vegetation and soil) is a major reservoir of carbon that acts as a net annual sink of atmospheric $CO_2$, therefore modulating the climate system (Heimann and Reichstein, 2008; Ciais et al., 2013) and are thought to offer a mitigation strategy to reduce global warming (Schimel et al., 2001). In European forests, 70% of the net land sink is sequestered in plants as woody biomass increments and 30% is sequestered in soils (Luyssaert et al., 2010). However, large uncertainty remains concerning the drivers and future of the soil organic carbon (Luyssaert et al., 2010). Therefore, investigating the mechanisms that impact storage and export of soil carbon from forest ecosystems is of first interest in both ecosystem and climate researches.

Streams and small rivers are important links between terrestrial and aquatic ecosystems because they receive inputs of carbon from land and then transform these materials at the land-stream interface and in stream channels, as water flows to larger rivers (McClain et al., 2003; Raymond et al., 2013). The carbon dynamics in forest stream ecosystems results from the interaction in soils between biological activity, weathering, retention mechanisms, water infiltration and drainage (Jones and Mulholland, 1998; Shibata et al., 2001; Kawasaki et al., 2005). Indeed, biogeochemical cycling within and across the terrestrial–aquatic interface is dynamically linked to the water cycle (Johnson et al., 2006; Battin et al., 2009), because dissolved carbon is primarily mobilized and transported by the movement of water (Hope et al., 1994; Hagerdon et al., 2000; Kawasaki et al., 2005). Furthermore, numerous works in different environments came to the conclusion that streams and small rivers are hotspots of $CO_2$ degassing (Johnson et al., 2008; Butman and Raymond, 2011; Polsenaere and Abril, 2012; Wallin et al., 2013; Kokic et al., 2015). In small stream, the $CO_2$ degassing flux mostly results from inputs of groundwater enriched in $CO_2$ (Hotchkiss et al., 2015), which comes from plant roots respiration and from microbial respiration in soils and groundwater.

The quantification of dissolved carbon fluxes transported by water from terrestrial to aquatic environments is fundamental to resolve the carbon balance at the catchment scale (Billett et al., 2004; Shibata et al., 2005; Jonsson et al., 2007; Kindler et al., 2011; Magin et al., 2017). Leaching of carbon from terrestrial ecosystems to streams could potentially represent up to 160% of the Net Ecosystem Exchange (NEE) in a Scotland peat catchment (Billett et al., 2004), 6% in a Sweden boreal catchment dominated by coniferous (Jonsson et al., 2007), on average 6% in five forest plots across Europe (Kindler et al., 2011), 2% in a Japanese temperate catchment dominated by deciduous forest (Shibata et al., 2005) or 2.7% of the Net Primary Production in different woody and tilled subcatchments across the southwest Germany (Magin et al., 2017). Such large variations in carbon export rates are not well understood and it is therefore important to extend this investigation to other landscapes and climatic

zones. More studies focused on the processes that govern the mobilization of soil carbon to surface

waters are necessary to improve and predict carbon budgets in terrestrial ecosystems.

Some authors reported high concentrations of dissolved inorganic carbon (DIC) (Kawasaki et al., 2005;

Venkiteswaran et al., 2014) and dissolved organic carbon (DOC) (Artinger et al., 2000; Baker et al.,

2000) in forest-dominated groundwater (i.e., in the saturated zone of the soil). However, estimations of

terrestrial carbon leaching from direct simultaneous measurements in groundwater and streams are

scarce. These studies are generally restricted to submarine and coastal environments (Santos et al.,

2012; Atkins et al., 2013; Sadat-Noori et al., 2016) and boreal lakes (Einarsdottir et al., 2017), but rarely

streams. The few studies that estimate exports of carbon from forested landscapes to streams are

generally based: (i) on carbon observations in soil water (i.e., in the unsaturated zone of the soil)

combined with soil water model that simulates the volume of soil water leached to streams (Öquist et

al., 2009; Kindler et al., 2011; Leith et al., 2015), (ii) on differences in the dissolved carbon flux

between upper and lower stream reaches combined with stream discharge (Shibata et al., 2001; Dawson

et al., 2002; Billett et al., 2004; Shibata et al., 2005; Olefeldt et al., 2013), or (iii) as described by the

active pipe concept (Cole et al., 2007), as the sum of the three major riverine carbon fluxes: $CO_2$

degassing, organic carbon burial in sediments and carbon export downstream (Jonsson et al., 2007).

These studies do not provide a complete understanding of the link between carbon hydrological export

and the physicochemical and biological processes occurring in soils and groundwater. In addition, the

approaches based on only stream sampling may miss part of the DIC export flux as excess $CO_2$ that

might rapidly degas upstream of the sampling points (Venkiteswaran et al., 2014).

In this study, we instrumented a temperate watershed that offers the convenience of a homogeneous

lithology (permeable sandy soil), vegetation (pine forest), topography (very flat coastal plain), as well as

a simple hydrological functioning exclusively as shallow groundwater drainage. This simple

configuration with no surface runoff allows us to identify what are the main factors that control the

DIC/DOC leaching to streams, the DIC:DOC ratio in groundwater and streams, and their variation in space and over time. At the plot scale, we relate DIC and DOC temporal dynamics in groundwater with hydrology and metabolic activity of the forest ecosystem. At the watershed scale, we quantify DIC and DOC transfers through the groundwater-stream interface, and we describe the fate of this carbon in first-order streams

## 2. Materials and Methods

### 2.1. Study site

The Leyre watershed (2,100 km²) is located in the southwest of France, in the "Landes de Gascogne" area (Fig. 1). The landscape is a very flat coastal plain with a mean slope lower than 0.125% (generally NW-SE) (Jolivet et al., 2007), but with local gentle slopes (notably near some streams). The mean altitude is lower than 50 m (Fig. 1) (Jolivet et al., 2007). The lithology is relatively homogeneous and composed of sandy permeable surface layers dating from the Plio-Quaternary period (Legigan, 1979; Bertran et al., 2009, 2011).

The podzolic soil is characterized by a low pH (4), low nutrient availability, and high organic carbon content that can reach 55 g per kg of soil (Augusto et al., 2010) . Three types of podzols are present: wet Landes (humic podzol), mesophyllous Landes (duric podzol) and dry Landes (loose podzol) that represents respectively 47%, 36% and 17%, of the watershed area (Jolivet et al., 2007; Augusto et al., 2010). Moreover, there is a gradient of soil carbon content from dry Landes (C = 6 to 17 kg m$^{-2}$) to mesophyllous Landes (C = 13 to 30 kg m$^{-2}$) and wet Landes (C = 15 to 30 kg m$^{-2}$) (Augusto et al., 2010). In the dry Landes of the upper parts of the watershed, the water table is always more than 2 meter deep. In the wet Landes of the lower parts, and in the vast interfluves, the groundwater is found near the soil surface in winter (0.0-0.5 m depth) and generally remains 1.0-1.5 meter deep in summer. The mesophyllous Landes corresponds to the intermediate situation (Augusto et al., 2006)

The region was a vast wetland until the 19[th] century, when a wide forest of maritime pine (pinus pinaster) was sown, following landscape drainage in 1850. Currently, the catchment is occupied mainly by pine forest (approximately 80%), with a modest proportion of croplands (approximately 15%) (Jolivet et al., 2007). The typical rotation period of pine forest is ~40 years, ending in clear-cutting, tilling and re-planting (Kowalski et al., 2003). The climate is oceanic with a mean annual air temperature of 13°C and a mean annual precipitation of 930 mm (Moreaux et al., 2011). Moreover, the average annual evapotranspiration of maritime pine is in the range of 234-570 mm (Govind et al., 2012). Owing to the low slope and the high permeability of the soil (hydraulic conductivity is approximately 40 cm h$^{-1}$, Corbier et al., 2010), the infiltration of rain water is fast (55 cm h$^{-1}$ on average, Vernier and Castro, 2010) and surface runoff does not occur; as the excess of rainfall percolates into the soil and recharges the shallow groundwater, causing the water table to rise. Moreover, very low content in feldspars and allover clay minerals in the sandy podzols induce a low water soil retention (Augusto et al., 2010). The superficial sandy soil contains a free and continuous water table strongly interconnected with the superficial river network; drainage is also facilitated by a dense network of drainage ditches, built in the 19[th] century, and currently maintained by forest managers in order to optimize tree growth (Thivolle-Cazat and Najar, 2001). In this study, we sampled first-order streams defined as streams and ditches with no tributaries and/or being seasonally dry.

## 2.2. Eddy covariance measurements at the forest plot scale

To quantify exchanges of carbon and water between the atmosphere and the pine forest plot, we used the site of Bilos (Fig. 1) (0.6 km², 44°29'38.08''N, 0°57'21.9''W, altitude: 40 m), as part of the ICOS research infrastructure (http://icos-ri.eu). In December 1999, the 50 years old pine forests was clear-cut (Kowalski et al., 2003). The site was ploughed to 30 cm depth and fertilized with 60 kg of $P_2O_5$ per ha in 2001 (Moreaux et al., 2011). In November 2004, the site was divided into two parts, which were seeded with maritime pine (pinus pinaster) with a 1-year lag, in 2004 and 2005, respectively,(Moreaux

et al., 2011). The forest plot was thus 10- and 11-year old during our sampling. The site was equipped with an eddy covariance measurement system soon after clear-cutting, and the system has been maintained since. The eddy covariance technique allows us to determine continuously the exchange between the ecosystem and the atmosphere of sensible heat, $CO_2$ and $H_2O$ by measuring the turbulent-scale covariance between vertical wind velocity and the scalar concentration of sensible heat, $CO_2$ and $H_2O$.

Wind velocity, temperature and $CO_2$/water vapor fluctuations were measured with, respectively, a sonic anemometer (model R3, Gill instruments Lymington, UK) and an open path dual $CO_2/H_2O$ infrared gas analyzer (model Li7500, LiCor, Lincoln, USA) at the top of a 9.6 m tower (01/01/2014 to 10/05/2014) and with another sonic anemometer (model HS50, Gill instruments) and an enclosed dual $CO_2/H_2O$ infrared gas analyzer (model Li7200, LiCor ©) at the top of a 15 m tower (09/07/14 to 31/12/2015). Consequently, there were no eddy covariance measurements available between 11/05/2014 and 08/07/2014 and thus between these two dates the latent heat fluxes were determined following the procedure of Thornthwaite (1948).

Raw data were processed following a standard methodology (Aubinet et al., 1999). The post-processing software EddyPro v6.0 (www.licor.com) was used to treat raw data and compute average fluxes (30 min period) by applying the following steps: (1) spike removal in anemometer or gas analyzer data by statistical analysis, (2) coordinating rotation to align coordinate system with the stream lines of the 30 min averages, (3) block average detrending of sonic temperature, $H_2O$ and $CO_2$ channels (4) determining time lag values for $H_2O$ and $CO_2$ channels using a cross-correlation procedure, (5) computing mean values, turbulent fluxes and characteristic parameters, and (6) spectral corrections (Ibrom et al., 2007). Thereafter, $CO_2$ and $H_2O$ fluxes were filtered in order to remove points corresponding to technical problems, meteorological conditions not satisfying eddy correlation theory or data out of realistic bounds. Different statistical tests were applied for this filtering: stationarity and

turbulent conditions were tested with the steady state test and the turbulence characteristic test recommended by Kaimal and Finnigan (1994) and Foken and Wichura (1996). Only values of $CO_2$ and $H_2O$ fluxes that pass all the filters were retained. Then, missing values of $CO_2$ and $H_2O$ fluxes were gap-filled. The NEE of $CO_2$ was partitioned into 2 components, Gross Primary Production (GPP) and Ecosytem respiration ($R_{eco}$) with the R package Reddyproc (version 0.8-2) applying the following steps (Reichstein et al 2005).

(i) during nighttime GPP = 0 so NEE = $R_{eco}$; (ii) statistical regression between $R_{eco}$ and night air temperature and meteorological conditions is adjusted with a Arrhenius type equation (Lloyd and Taylor, 1994); (iii) day-time $R_{eco}$ is obtained by extrapolating night-time fluxes using the temperature response; (iv) GPP is calculated as the difference between daytime NEE and $R_{eco}$, additional checks are performed to avoid unrealistic values of GPP. Finally, a positive NEE indicates an upward flux whereas a negative NEE indicates a downward flux, GPP is positive or zero and $R_{eco}$ is positive. NEE = $R_{eco}$.- GPP.

## 2.3. Groundwater and surface water monitoring

To compare groundwater carbon dynamics at both the plot and at the watershed scales, we selected 3 piezometers in different forest types (Fig. 1). According to the depth and amplitude of the water table, the three piezometers were representative of dry Landes (Piezometer 2), mesophyllous Landes (Piezometer 3) and a situation between mesophyllous and wet Landes (Piezometer Bilos). Moreover, the piezometer 2 is located in a riparian mixed pine and oak forest near a first-order stream whereas the piezometer 3 is located in another pine forest (approximately same age as Bilos pine forest).We also selected six first-order streams whose watersheds were dominated largely by pine forest (~90%) which limit biogeochemical signal from crops. Shallow groundwater and stream waters were sampled for partial pressure of $CO_2$ ($pCO_2$), total alkalinity and DOC with approximately a monthly time intervals (Tab. S1).

## 2.4. Chemical analysis

We measured the $pCO_2$ directly in the field and total alkalinity and DOC back in the laboratory. The $pCO_2$ in the groundwater and streams was measured directly using an equilibrator (Frankignoulle and Borges, 2001; Polsenaere et al., 2013). This equilibrator was connected to an Infra-Red Gas Analyzer (LI-COR®, LI-820), which was calibrated one day before sampling, on two linear segments because of its non-linear response in the range of observed $pCO_2$ values (0–90,000 ppmv). This non-linearity was due to saturation of the infrared cell at $pCO_2$ values above 20,000 ppmv. We used certified standards (Air Liquide™ France) of 2,079±42; 19,500±390 and 90,200±1,800 ppmv, as well as nitrogen flowing through soda lime for zero. For the first linear segment [0-20,000 ppmv], which corresponded to the river waters, we set the zero and we spanned the LI-COR at 19,500 ppmv, and then checked for linearity at 2,042 ppmv. For the second segment [20,000-90,000 ppmv], which corresponded to the sampled groundwater, we measured the response of the LICOR with the standard at 90,000 ppmv, and used this measured value to make a post correction of the measured value in the field. Before sampling, the groundwater was pumped from the piezometer during the time necessary to obtain stable readings of temperature, pH, electrical conductivity and dissolved oxygen concentration.

Total alkalinity was analyzed on filtered samples by automated electro-titration on 50 mL filtered samples with 0.1N HCl as the titrant. The equivalence point was determined from pH between 4 and 3 with the Gran method (Gran, 1952). The precision based on replicate analyses was better than ± 5 µM. For samples with a very low pH (<4.5), we bubbled the water with atmospheric air in order to degas the $CO_2$. Consequently, the initial pH increased above the value of 5, and total alkalinity titration could be performed (Abril et al., 2015). We calculated DIC from $pCO_2$, total alkalinity and temperature measurements using carbonic acid dissociation constants of Millero (1979) and the $CO_2$ solubility from Weiss (1974) as implemented in the $CO_2SYS$ programme (Lewis et al., 1998). Contrary to the $pCO_2$ calculation from pH and total alkalinity (Abril et al., 2015), the DIC calculation from measured $pCO_2$

and total alkalinity was weakly affected by the presence of organic alkalinity, because 80±20 % of DIC in our samples was dissolved $CO_2$. The DOC samples were obtained after filtration, in the field through pre-combusted GF/F filters (porosity of 0.7 µm). The samples were acidified with 50 µL of HCl 37% to reach pH 2 and stored in pre-combusted Pyrex 25 mL vials at 4 °C in the dark before analysis. The

DOC concentrations were measured with a SHIMADZU TOC 500 analyzer (in TOC-IC mode), with repeatability better than 0.1 mg $L^{-1}$.

## 2.5. Hydrological monitoring

The precipitation was measured continuously at the Bilos plot using automatic rain gauges with a 30 minutes integration: one Young EML SBS 500 (EML, North Shields, UK) was located in a small clear-

cut at 3 m above ground from 01/01/2014 to 10/05/2014 and one electronic gravimetric heated precipitation gauge TRwS (MPS system; Bratislava, Slovakia) was located at the top of the canopy on a 6 m tower, from 01/07/2014 to 31/12/2015. Hence, between 11/05/2014 and 31/06/2014, none precipitation measurements were available at the Bilos site. Thus, during this period, we used data from Meteo France © station at Belin-Béliet (approximately 30 km from the Bilos site). The precipitation

measurements were also checked weekly in the field with manual reports.

The groundwater table depth was measured continuously at the Bilos plot using high performance level pressure sensors (PDCR/PTX 1830, Druck and CS451451, Campbell Scientific) in one piezometer located amid the Bilos site. The pressure measurements were fully compensated for temperature and air

pressure fluctuations. The measurements were obtained at 60-seconds intervals and integrated on 30-minutes period. They were checked with manual probe weekly. The groundwater table depth was also measured punctually with a manual piezometric probe in the piezometer 2 and 3 before each groundwater sampling.

Our study benefited from four calibrated gauging stations of the DIREN (French water survey agency), with a daily temporal resolution, located on two second-order streams (Bourron and Grand Arriou rivers), one third-order stream (Petite Leyre river) and one fourth-order stream (Grande Leyre river) (Fig.1). We also performed additional discharge measurements in first-order streams (Fig. 1). For each stream order, we calculated with a daily temporal resolution for a two-year period the drainage (i.e.,

discharge divided by the corresponding catchment area, in $m^3$ $km^{-2}$ $d^{-1}$ or in mm $d^{-1}$) (Deirmendjian and Abril, in revision). We then determined the increase of drainage between two streams of successive orders. Because of the specific characteristics of the Leyre watershed with no surface runoff, we observed a regular increase in drainage values between two streams of successive orders. In addition, the proportion of additional drainage occurring in each stream order was relatively constant temporally.

Our analysis based on daily discharge monitoring in second-, third-, and fourth-order streams and seasonal gauging of first-order streams revealed that monthly drainage values in first order streams were on average 2.3 times lower than that measured in fourth order stream and allowed us to reconstruct robust monthly drainage values in first order streams (Deirmendjian and Abril, in revision). We wrote the water mass balance equation at the Bilos forest plot as follows:

$$P = D + ETR + GWS + \Delta S \qquad \text{(Eq. 1)}$$

where P, D, ETR, GWS and $\Delta S$ were respectively, precipitation, drainage, evapotranspiration, groundwater storage and change of soil water content in the unsaturated zone, all expressed in mm $d^{-1}$. P was the cumulative precipitation measured over a given period t at the Bilos site. D was the drainage at

270 the Bilos site deduced from daily observation at four gauging stations and the hydrological model (Deirmendjian and Abril, in revision). ETR was the cumulative evapotranspiration obtained from eddy covariance measurements of latent heat fluxes over a period t at the Bilos site. GWS was calculated as the net change in water table depth over the period t times the e soil effective porosity at the Bilos site of 0.2 ((Augusto et al., 2010; Moreaux et al., 2011). Finally, none reliable measurements of soil water

content were available and the $\Delta S$ term being likely small he variation of soil water content in the unsaturated zone was neglected in the water mass balance.

## 2.6. Carbon stocks in groundwater, exports to streams and degassing to the atmosphere

We calculated four different terms that describe the dynamics of carbon at the Bilos plot: the stocks of DIC ($DIC_{stock}$) and DOC ($DOC_{stock}$) in groundwater and the exports of DIC ($DIC_{export}$) and DOC
($DOC_{export}$) from groundwater to first-order streams; all integrated between two sampling dates (Tab. S2). Because we do not know the total height of the permeable surface soil layer in the piezometer 2 and 3, we calculated the stocks of carbon in the groundwater only at the Bilos site. However, in order to account for spatial differences between the dry, mesophyllous and wet Landes, specific DIC and DOC exports were calculated for the three study sites piezometers. We wrote:

$$DIC_{stock} = (S_i + S_f) / 2 = (DIC_i \times V_i + DIC_f \times V_f) / 2 \qquad \text{(Eq. 2)}$$

where $DIC_{stock}$ was the mean stock of DIC in groundwater between two sampling dates in mmol m². $S_f$ and $S_i$ were the final and the initial stock of DIC in groundwater in mmol m$^{-2}$. $DIC_i$ and $DIC_f$ were the initial and the final concentration of DIC in groundwater in mmol m$^{-3}$, respectively. $V_i$ and $V_f$ were the
290 initial and the final volume of groundwater in m$^3$ m$^{-2}$. The volume of groundwater (V) was calculated as follows:

$$V = (h + H) \times \Phi_{effective} \qquad \text{(Eq. 3)}$$

where h and H (H is negative), were respectively the total height of the permeable surface layer (equals
to 10 m, Corbier et al., 2010) and the height of groundwater table. $\Phi_{effective}$ was the effective porosity of the soil and it was equal to 0.2. Export of DIC in first-order streams through drainage of shallow groundwater was calculated from discharge and concentration as follows:

$$DIC_{export} = D \times (DIC_i + DIC_f) / 2 \qquad \text{(Eq. 4)}$$

where D was the mean drainage of shallow groundwater by first-order streams between the initial and

the final sampling dates in m $d^{-1}$. $DIC_i$ and $DIC_f$ were the initial and the final concentration of DIC in

groundwater in mmol $m^{-3}$. We calculated $DOC_{stock}$ and $DOC_{export}$ as the same manner as $DIC_{stock}$ and

$DIC_{export}$. In addition, we also calculated the DIC exported from first-order streams to second-order

streams by replacing in the equation 4 the concentrations of carbon in the groundwater by carbon

concentrations in first-order streams. Between two sampling dates, the degassing of $CO_2$ in first-order

streams could thus be obtained from the difference between the DIC exported from groundwater and

from first-order streams.

## 3. Results

### 3.1. Hydrological parameters and water mass balance

Water mass balance at the Bilos site was calculated on a monthly basis over a two-year period (2014-

2015) (Tab. 1; Figs. 2c, 3). Monthly precipitation on the one hand and the sum of evapotranspiration,

groundwater storage and drainage on the other hand closely followed the 1:1 line (Fig. 3), showing the

consistency of the water mass balance estimated with different techniques and independent devices,

even with a monthly temporal resolution not sufficient to account for very sudden processes. During the

315 years 2014 and 2015, we could define four different hydrological periods that were high flow, growing

season, late summer and early winter periods (Fig. 2). High flow periods were characterized by two

relatively short flood events in Jan. 2014-Mar. 2014 (peak of 120 $m^3$ $s^{-1}$) and in Feb. 2015-Mar. 2015

(peak of 80 $m^3$ $s^{-1}$), high drainage values (maximum of 1.9 mm $d^{-1}$ in Feb. 2014) and a water table close

to the soil surface (Tab. 1; Figs. 2a-c). These short periods of high flow in winter were followed by the

320 forest-growing season in spring and summer in May. 2014-Aug. 2014 and Apr. 2015-Aug. 2015

characterized by highest GPP and $R_{eco}$ (maximum of 880 and 660 mmol $m^{-2}$ $d^{-1}$, respectively, in May

2015) and highest evapotranspiration (maximum of 5.3 mm $d^{-1}$ in Apr. 2014); during this growing

period, the groundwater table decreased and groundwater storage was negative (Tabs. 1, 3; Fig. 2).

Growing season periods were followed by late summer periods that were characterized by low precipitations (miminum of 0.2 mm $d^{-1}$ in Sep. 2014), and the lowest groundwater table depth in Sept. 2014-Oct. 2014 and in Sep. 2015-Oct-2015 (Tab. 1; Fig. 2a-c). Late summer periods were followed by early winter periods that were associated with heavy precipitations (maximum of 4.7 mm $d^{-1}$ in Nov. 2014) and rising groundwater table (positive groundwater storage) in Nov. 2014-Jan. 2015 and in Nov. 2015-Dec.2015 (Tab. 1; Figs. 2a-c). We considered that, growing season, late summer and early winter periods, merged together, represented periods of base flow.

Periods of groundwater discharge with negative groundwater storage (Feb. 2014-Sep. 2014 and Mar. 2015-Aug. 2015) were characterized by evapotranspiration higher than precipitations (Figs. 2a-c). Conversely, periods of groundwater recharge with positive groundwater storage (Oct. 2014-Feb. 2015 and Sep. 2015-Dec. 2015) were characterized by precipitations higher than evapotranspiration (Figs. 2a-c). Consequently, at the plot scale, significant correlations between groundwater storage and precipitations and between groundwater storage and evapotranspiration were observed (Tab. 2), attesting that evapotranspiration and precipitations played a significant role in the groundwater storage.

### 3.2. Net Ecosystem exchange of $CO_2$ in the forest plot (Bilos plot)

GPP, $R_{eco}$ and NEE exhibited a strong seasonal variability (Tab. 3; Fig. 2b). GPP. $R_{eco}$ and NEE were respectively 400±220 mmol $m^{-2}$ $d^{-1}$, 310±150 mmol $m^{-2}$ $d^{-1}$ and -90±110 mmol $m^{-2}$ $d^{-1}$ throughout the years 2014 and 2015 (we excluded the 16/05/14-07/07/14 period when none eddy covariance measurements were available), equivalent to 1,750±960; 1,360±660 and 390±480 g C $m^{-2}$ $yr^{-1}$ (Tab. 3; Fig. 2b). These results were close from Moreaux et al (2011) estimates of 1,720; 1,480 and 340 g C $m^{-2}$ $yr^{-1}$ respectively, as measured at a younger forest stage in the same forest plot. GPP increased from early winter (210±30 mmol $m^{-2}$ $d^{-1}$) to growing season (640±150 mmol $m^{-2}$ $d^{-1}$) periods (Tab. 3; Fig. 2b). $R_{eco}$ followed the same temporal trend (Tab. 3; Fig. 2b). During late summer and early winter periods, NEE could be positive ($R_{eco}$>GPP), meaning that the pine forest ecosystems had switched from

autotrophic to heterotrophic metabolism, notably in Oct, Nov and Dec. 2014 (Tab. 3; Fig. 2b). NEE was

always negative ($R_{eco}$<GPP) during high flow and growing season periods, except in Jul. 2015, probably

as a consequence of temporary low precipitation (Tab. 3; Figs. 2b-c).

### 3.3. Dissolved carbon evolution in shallow groundwater

In shallow groundwater, total alkalinity was low and originated from slow weathering of silicate

minerals with vegetation-derived $CO_2$ (Polsenaere and Abril, 2012). The mean proportion of total

alkalinity in the DIC pool in shallow groundwater was 5%, the large majority of the DIC being

composed of dissolved $CO_2$ resulting from microbial and plant root respiration in the soil. Although the

sampling frequency was monthly or more, it allowed to detect significant changes in the groundwater

DIC and DOC concentrations, consistent from one year to another at Bilos site and from one site to

another during the second hydrological year of the study (Figs. 4, 5). One first and relevant key result

was the opposite temporal evolution of DIC and DOC concentrations in groundwater with water table

depth (Tab. 2; Figs. 4, 5). Indeed, DIC and DOC concentrations in groundwater exhibited strong

temporal variations in relation with the hydrological cycle (Tab. 4; Figs. 4, 5). On the one hand, during

high flow and growing season periods of 2014, the increase of DIC in Bilos groundwater (570 to 3,030

µmol L$^{-1}$) was associated with a fast decrease of DOC in Bilos groundwater (3,625 to 950 µmol L$^{-1}$), in

parallel with a decline in the water table (Fig. 5). In 2015, the same temporal trend was observed at the

same period, but with a lesser extent (Fig. 5). On the other hand, during period of late summer, the

increase of DIC concentrations in Bilos groundwater (2,700 to 5,400 µmol L$^{-1}$) was this time not related

with any decrease of DOC concentrations in groundwater (Figs. 5b-c). This maximum of DIC

concentrations in groundwater corresponded of late summer period when the overlying forest ecosystem

had switched from autotrophic to heterotopic metabolism (Figs. 2b, 5). During early winter and high

flow periods, DIC concentrations in Bilos groundwater decreased from 4,000 µmol L$^{-1}$ (Nov. 2014) to

1,700 µmol L$^{-1}$ (Mar. 2015), in parallel with a rise in the water table (Figs. 5a-b). Concomitantly, a fast

increase in DOC concentrations from 670 to 3,600 µmol L$^{-1}$ occurred in Bilos groundwater between the same time periods (Fig. 5a-c).

The DIC concentrations in the three sampled piezometers exhibited a modest spatial heterogeneity (Tab. 4; Fig. 5b). DIC concentrations were low (e.g., 570 µmol L$^{-1}$ in the Bilos piezometer in Feb. 2014) during periods of high flow and were high (e.g., 5,370 µmol L$^{-1}$ in the Bilos piezometer in Sep. 2014) during period of late summer (Tab. 4; Figs. 5a-b). In contrast to DIC, the DOC concentrations exhibited

a significant spatial heterogeneity, particularly during high flow periods (Tab. 4; Fig. 5). During these periods of high flow, DOC concentrations were higher in the Bilos piezometer (3,800±200 µmol L$^{-1}$) than in the piezometer 2 (280 µmol L$^{-1}$) and 3 (1,500 µmol L$^{-1}$) (Tab. 4; Fig. 5a-c). During the other hydrological periods (periods of base flow), DOC concentrations in the piezometer 2 were still lower than the two other piezometers (Bilos & 3) (Tab. 5). However, during periods of base flow,

groundwater DOC concentrations in the three sampled sites remained more or less constant (Tab. 4; Figs. 5a-c).

**3.4. Dissolved carbon evolution in first-order streams**

In first-order streams, the DIC concentrations exhibited smaller temporal variations and significantly lower values than in groundwater, attesting that degassing occurred at the groundwater-stream interface

(Tab. 4; Fig. 5b). In contrast to DIC, the DOC concentrations in first-order streams were of the same order of magnitude than in the piezometer 2 (dry Landes) and significantly lower than in the two other piezometers (wet to mesophyllous Landes), in particular during periods of high flow (Tab. 4; Fig. 5c) . As in groundwater, DOC and DIC concentrations in first-order streams were significantly anti-correlated (Tab. 2), suggesting that carbon dynamics in first-order streams was mostly impacted by

groundwater inputs.

### 3.5. Carbon stocks in groundwater and exports to streams

At the Bilos site, the stocks of DIC and DOC in groundwater followed the same temporal trend than DIC and DOC concentrations (Figs. 5, 6). The stock of DIC increased from high flow (1,140 mmol m$^{-2}$ the 12/02/14) to late summer (8,700 mmol m$^{-2}$ the 24/09/2014) periods, whereas at the same time intervals, the stock of DOC decreased from 7,240 mmol m$^{-2}$ to 780 mmol m$^{-2}$ (Fig. 6). Furthermore, between 12/02/2014 and 16/05/2014 (95 days), we observed an increase of 4,500 mmol m$^{-2}$ in DIC stocks very close to the decrease of DOC stocks of 5,500 mmol m$^{-2}$. This suggests that during the following months after the DOC peak in groundwater at high flow period, DOC is degraded to DIC within the groundwater itself. During this period, the degradation rate of DOC in the groundwater could be estimated at approximately 60 mmol m$^{-2}$ d$^{-1}$.

The export of DOC occurred in majority during high flow periods (e.g., 90% of the total DOC export in Bilos plot occurred during high flow periods), for each sampled groundwater (Tab. 5). During high flow periods, the groundwater DOC concentrations and exports exhibited an important spatial heterogeneity at the three sampled site (Tab. 5). During these periods of high flow, DOC export was higher in the Bilos piezometer (3.4±1.1 mmol m$^{-2}$ d$^{-1}$) than in the piezometer 2 (0.4±0.02 mmol m$^{-2}$ d$^{-1}$) and in the piezometer 3 (1.5±0.2 mmol m$^{-2}$ d$^{-1}$) (Tab. 5). These contrasts in DOC exports were related to the water table depth and amplitude (Fig. 4), and the gradient in soil carbon between the different podzols. In contrast to DOC exports, approximately the same quantity of DIC was exported during high flow periods (e.g., 50% of the total DIC export in Bilos plot occurred during HF period) than during the other hydrological periods, for each sampled groundwater (Tab. 5). Groundwater DIC exports, exhibited a smaller spatial heterogeneity than DOC exports although DOC and DIC concentrations showed opposite seasonal trend in groundwater (Tabs. 4, 5, Figs. 5b-c); the time-integrated value of carbon export for the sampling period was 0.9±0.5 mmol m$^{-2}$ d$^{-1}$ (3.9±2.2 g C m$^{-2}$ yr$^{-1}$) for DIC and 0.7±0.7 mmol m$^{-2}$ d$^{-1}$ (3.1±3.1 g C m$^{-2}$ yr$^{-1}$) for DOC (Tab. 5). As drainage of groundwater was the only

hydrological pathway in the Leyre watershed, terrestrial carbon leaching to streams was estimated to be $1.6\pm0.9$ mmol m$^{-2}$ d$^{-1}$ ($7.0\pm3.9$ g C m$^{-2}$ yr$^{-1}$).

### 3.6. Degassing in first-order streams

Degassing in first-order streams was $0.7\pm0.5$ mmol m$^{-2}$ d$^{-1}$ ($3.1\pm2.2$ g C m$^{-2}$ yr$^{-1}$) throughout the sampling period (Tab. 5). Degassing was more important during periods of high flow than during the other hydrological periods (Tab. 5). In addition, degassing in first-order stream was positively correlated to the export of DIC (Tab. 2), revealing that degassing was mostly impacted by groundwater inputs. Over a hydrological year, 75% of the DIC exported from the Leyre watershed based on the three groundwater sampling sites, almost immediately returned in the atmosphere through $CO_2$ degassing in first-order streams (Tab. 5).

## 4. Discussion

### 4.1. Water mass balance and the role of groundwater in the hydrological carbon export

Our hydrological dataset monitored continuously during 18 months allows us to separate the water budget in four terms at the monthly timescale (Tab. 1; Figs. 2c, 3). The water budget at the Bilos plot was primarily impacted by precipitation and evapotranspiration (Tab. 1; Fig. 2c). The transfer of precipitation to rivers involves temporary water storage in groundwater (Alley et al., 2002; Oki and Kanae, 2006). The lag time between precipitation and groundwater storage was short at our study site, as attested by the significant correlation between these two parameters (Tab. 2). Thus, when precipitations are high (during early winter and high flow periods), water infiltration in the sandy podzols is faster than water capture by vegetation. Consequently, rainwater infiltration rapidly causes the water table to rise and thus increases the groundwater storage (Figs. 2a-c). This fast infiltration is due to the sandy permeable texture of soils with a low water soil retention (Augusto et al., 2010;Vernier and Castro, 2010).

The evapotranspiration was high during growing season and late summer periods when the precipitations were low (Tab. 1; Fig. 2c). For that reason, the groundwater storage decreases with increasing evapotranspiration (Tab. 2; Fig. 2c), revealing that soil water uptake by the pine trees directly lowers the water table. Soil water retention properties usually vary with depth and thus soil water uptake by plant roots generally occurs from areas in the soil with the highest water potential (Warren et al., 2005; Domec et al., 2010). Previous studies suggest that the ordinary soil depth at which most water is taken up in pines is usually 30–40 cm (Querejeta et al., 2001; Klein et al., 2014) where nutrient concentrations are also the highest (Achat et al., 2008). In an experimental Scots Pine plot in a flat and sandy area of Belgium, similar as our study site, Vincke and Thiry (2008) reported that water table uptake could contribute to 60% of the evapotranspiration thanks to capillary rise from the groundwater up to the rooted soil layers. To the contrary to the pine trees, direct groundwater table uptake has been observed for deciduous trees in a flat and sandy area of Portugal (Mendes et al., 2016), a process that occurred through a dimorphic root system which allows the access and use of groundwater resources (David et al., 2013) in particular during drought period (Del Castillo et al., 2016). Evapotranspiration strongly controls the groundwater storage in pine forests and, as a result, water table generally rises after clear-cut (Bosch and Hewlett, 1982; Sun et al., 2000; Xu et al., 2002). At our study site, drainage also significantly increased after wood harvesting due to reduced evapotranspiration, (Kowalski et al., 2003; Loustau and Guillot, 2009). Indeed, the network of drainage ditches created by foresters evacuates very rapidly the water in excess when the groundwater level rises (Thivolle-Cazat and Najar, 2001). Since most pine roots are located in the first meter of the soil to avoid winter anoxia caused by rising water table (Bakker et al., 2006, 2009), the pine trees did not exhibit any transpiration reduction when the groundwater level is high (Figs. 2a-c; Loustau et al., 1990).

We observed a lag time between groundwater storage and drainage at our study site (Figs. 2a-c, 3), confirmed by the non-significant correlation between these two parameters (Tab. 2). This lag of 2-3

470   months was due to the time necessary for water to travel through the soil depending on the spatial

temporal gradient of hydraulic head, hydraulic conductivity, and porosity of the system (Alley et al.,

2002; Ahuja et al., 2010). At our study site, shallow groundwater acts as a buffer system, the drainage

being mostly controlled by water table depth and the capacity of the porous soil to store or export water

(Alley et al., 2002). Indeed, groundwater flow in a shallow sandy aquifer is largely controlled by the

drainage pattern of the streams and ditches, and thus by the water table depth and topography of the area

(Vissers and van der Perk 2008). At our study site, the buffer capacity of groundwater  explains why the

Leyre river discharge increased only in late winter, 2-3 months after the start of high precipitations and

rising water table (Figs. 2a-c). Sudden hydrological events are thus buffered by this temporary

groundwater storage in the porous soil. As a consequence, temporary groundwater storage mediates

almost all the carbon exports to the watershed. Moreover, storms would not have such crucial impact on

the way we estimate carbon exports from groundwater to first-order streams, based on monthly

sampling frequency. Indeed, with our monthly resolution we observed consistent seasonal effect of DIC

and DOC in shallow groundwater and streams (Fig. 5), representative for the different processes that

control carbon dynamics in groundwater. On the contrary, in steeper and less permeable catchments,

carbon exports are quickly affected by storms and pulsed hydrological events (e.g., Raymond and

Saiers, 2010; Wilson et al., 2013). Finally, the water mass balance at the Bilos plot scale being

consistent with drainage modeled at the watershed scale (Figs. 2c, 3), we used this drainage to estimate

carbon exports at the plot scale (Tab. 5).

**4.2. Soil carbon leaching to groundwater**

Dissolved carbon concentrations varied considerably in groundwater (Tab. 4; Figs. 4, 5) according to

seasonal changes in hydrology and forest metabolism and depending on the characteristics of the

sampling site. Because the sampling frequency was approximately one month, we may have lost some

short transitional periods significant for the annual carbon budget. This is most probable during the

short period of high flow, when DOC mobilization and export were the highest (Tab. 5). However, the

sampling frequency was sufficient to detect the major trends in groundwater DIC and DOC

concentrations, consistent from one hydrological cycle to another at the Bilos site and from one site to another during the second hydrological year of the study, although topographic differences explained spatial differences in DOC and DIC concentrations (Fig. 5). Thanks to the high permeability of the soil and the buffering capacity of the groundwater in response to hydrology, we could observe distinct
biogeochemical processes that govern carbon leaching throughout the hydrological cycle.

Dissolved organic matter generally includes a small proportion of low molecular weight compounds such as carbohydrates and amino acids and a larger proportion of complex, high molecular weight compounds (Evans et al., 2005; Kawasaki et al., 2005). Dissolved organic matter is often quantified by
its carbon content and referred to as DOC and nearly all DOC in soils come from photosynthesis (Bolan et al., 2011). Indeed, DOC in soils forests originates from throughfall and stemflow, leaf litter leaching, root exudation and decaying fine roots in soils (Bolan et al., 2011). However, a large fraction of DOC in soil solution is sorbed onto minerals and, before being exported to streams, DOC must be mobilized from the soil (Sanderman and Amundson, 2009). Surface precipitation has been described as an
important process that transports DOC downward from the topsoil to the saturated zone (Kawasaki et al., 2005; Shen et al., 2015). The transfer of DOC in groundwater also depends on the level of hydraulic connectivity between subsoils horizons and water table depth (Kalbitz et al., 2000). However, up to 90% of surface-derived DOC can be removed by re-adsorption to minerals, prior reaching the saturated zone (Shen et al., 2015). Furthermore, when sorptive retention of DOC occurs, it contributes to carbon
accumulation in subsoils due to the stabilization of organic matter against biological degradation (Kaiser and Guggenberger, 2000; Kalbitz and Kaiser, 2008). During base flow conditions, the DOC concentrations in groundwater were relatively stable at our study sites, even after rainy periods (Tab. 4; Figs. 2c, 5c), which suggests that soil DOC in upper horizons was not preferentially mobilized to groundwater by rainwater infiltration. Spatially, groundwater DOC was on average higher at the
mesophyllous to wet Landes station (Bilos and Piezometer 3), than at the dry Landes (Piezometer 2) during low water table periods (Tab. 4). Indeed, several studies have reported decreasing DOC

concentrations in groundwater in concurrence with increasing subsoil thickness and water table depth (Pabich et al., 2001; Datry et al., 2004; Goldscheider et al., 2006), with DOC concentrations at or close to zero reported in deep (> 1 km) and old groundwater (Pabich et al., 2001). At our study site, the

525 fraction of groundwater DOC that predominates at low water table was probably more recalcitrant, more stabilized and more aged than during high flow. Indeed, in forested watersheds, the [14]C age of groundwater DOC generally varies from old DOC at base flow to relatively modern DOC during high flow (Schiff et al 1997).

In the podzol soils of the Landes de Gascogne, the saturation of the superficial organic-rich horizon of the soil was necessary to generate very high DOC concentrations in the groundwater (Figs. 4, 5, 7). This suggests changes in the chemical conditions that altered the DOC retention capacity of the soil. In temperate forested ecosystems, leaching of DOC from subsoils is generally controlled by retention in the mineral B horizon of the soil with high content of extractable aluminum and iron oxides (Michalzik

et al 2001; Kindler et al 2011). In sandy podzols that contain almost no clay minerals (Augusto et al., 2010), DOC retention in soil is mainly controlled by DOC-metal complex (Lundström et al., 2000; Sauer et al., 2007). These Al-Fe oxides are considered as the most important sorbents for dissolved organic matter in soils (Kaiser et al., 1996). In podzols such as our study site, the content in Al-Fe oxides, and their degree of complexation by soil organic matter increases with depth (Ferro-Vázquez et

al., 2014; Achat et al., 2011). When the water table rises and reaches the organic-rich horizon of the soil, reducing conditions in the saturated soil will prevail. Indeed, we observed anoxic conditions in groundwater all year round at the Bilos site (data not shown). Under such reducing conditions in the saturated soil, dissolution of Fe oxides can occur, limiting the sorptive retention of DOC (Hagedorn et al., 2000; Camino-Serrano et al., 2014; Fang et al., 2016). DOC is then released to groundwater,

transported downward, partly retained in the mineral horizon of the soil and exported to streams. During these high flow periods, groundwater DOC peaked at a significantly higher value at the mesophyllous to wet Landes station (Bilos), than at the mesophyllous Landes (piezometer 3) and at the dry Landes

(Piezometer 2) (Tab. 4). This is a consequence of the water table depth and amplitude and the different carbon content in the superficial layers of the soil (Fig. 4). We calculated a stock of soil organic carbon in the 0-60 cm layer of 9.7 kg m$^{-2}$ at the Bilos plot (Trichet and Loustau, personal communication) whereas the stocks of DOC and DIC in Bilos groundwater were on a yearly average respectively 0.03 and 0.06 kg m$^{-2}$ (Fig. 6). As dissolved carbon in groundwater represents approximately 1% of the soil carbon, only a small part of the soil organic carbon content is leached into groundwater and potentially exported to streams.

The three months (Mar. 2014-May. 2014) following the flood peak of 2014, DOC concentrations and stocks in Bilos groundwater decreased regularly in parallel with an increase in DIC concentrations and stocks in groundwater (Figs. 5b-c, 6). The DOC degradation and DIC accumulation rates in Bilos groundwater were very similar and estimated of approximately 60 mmol m$^{-2}$ d$^{-1}$, or 6.5 mmol m$^{-3}$ d$^{-1}$. This DOC degradation occurred during decreasing water table periods although these periods are characterized with moderate groundwater temperature (<13°C). Moreover this DOC degradation rate is consistent with findings of Craft et al (2002) who reported respiration rates within the range of 3-100 mmol m$^{-3}$ d$^{-1}$ within a floodplain aquifer of a large gravel-bed river in north-western Montana in USA. As the same manner, in a semi-arid mountain catchment in New Mexico, Baker et al (2000) also observed that groundwater DOC peaked during periods of high flow and resulted in higher rates of heterotrophic metabolism, presumably because of the supply of labile DOC via more intense hydrologic connections between the soil and the groundwater. The bioavailability of groundwater DOC is related with the content of low molecular weight compounds, such as total dissolved amino acids, high molecular weight compounds, such as fulvic or humic acids are being more recalcitrant to decomposition by microbes (Baker et al., 2000; Shen et al., 2015). Our results suggest that DOC degradation within the groundwater occurred the following months after the mobilization of biodegradable DOC during high water table.

The increase of DIC concentrations in groundwater during late summer of 2014 (Sep-Oct. 2014) was
due to another process, this time not associated with any DOC degradation in groundwater (Figs. 5b-c).
At our study site, the late summer period, when the forest ecosystem is a net source of $CO_2$ for the
atmosphere (positive NEE), also corresponds to a maximum in $CO_2$ concentration in groundwater (Fig.
2b, 5b) and thus a maximum contribution of soil respiration to groundwater DIC. Transfer of $CO_2$ from
soil air to groundwater requires input of fluid, i.e., gas or water (Tsypin and Macpherson, 2012).
Typical pathways are downward $CO_2$ transport from soil in the dissolved (Kessler and Harvey, 2001) or
gaseous form (Appelo and Postma, 2004), upward flux of deep $CO_2$ of various origins through gas vents
(Chiodini et al., 1999) or leakage from adjacent aquifers. At our study site there is no evidence of deep
$CO_2$ source or leakage from adjacent aquifers (Bertran et al., 2011). In addition, during late dry summer
no rainy events occurred (Fig. 2c), and the high temperature observed during this period are favorable
for a high production of gaseous $CO_2$ in the unsaturated region of the soil which follows the Arrhenius
equation (Lloyd and Taylor, 1994; Reth et al., 2005). During high temperature periods in summer, the
amount of $CO_2$ in equilibrium with groundwater lower than in soil upper horizons favored a downward
flux of gaseous $CO_2$ (Tsypin and Macpherson, 2012), which suggests that soil $CO_2$ must have been
transported to groundwater in gaseous form by simple downward diffusion (Fig. 7). In a North
American tallgrass prairie resting on limestone, downward movement of $CO_2$ gas followed by
equilibration with groundwater at the water table was favorable during drought period whereas transport
of soil $CO_2$ in the dissolved form with diffuse flow of recharge water was the most effective during wet
periods (Tsypin and Macpherson, 2012). In temperate forested landscapes, other authors noticed that
during dry periods, a strong reduction in soil $CO_2$ flux to the atmosphere (upward diffusion) is
associated with a decline in soil water content that stresses roots and microorganisms (Davidson et al.,
1998; Epron et al., 1999). This suggests that the peak of groundwater $pCO_2$ observed in October (Fig.
5b) originates from soil $CO_2$ that was produced before, certainly during Jul-Aug when the temperature
was the highest and precipitations were sufficient to maintain a soil moisture that did not limit soil
respiration (Figs. 2b-c). The lag time of 2-3 months between the peak of groundwater $CO_2$ and soil $CO_2$

has been documented by Tsypin and Macpherson (2012) who concluded that it correspond to the travel time of soil-generated $CO_2$ to the water table. In the sandy podzols, during the drought period, the high porosity in the sandy soil may favor downward diffusion of $CO_2$ and its dissolution in groundwater. Thereafter, during early winter period, concentrations of DIC in groundwater decreased as a consequence of dilution with rainwater with low DIC content (Fig. 7).

## 4.3. Carbon transfer at the groundwater-stream-atmosphere interface

In the Leyre watershed, carbon exports are influenced with the soil types, which are characterized with a different water table depth and amplitude (Fig. 4), as well as a gradient of carbon content in the different soil types (Augusto et al., 2006). However, these last parameters have a stronger effect on the spatial heterogeneity of DOC exports than DIC exports (Tab. 5). Indeed, drainage and DOC concentrations in groundwater have a cumulative positive effect on DOC exports (Tabs 1, 2, 4, 5; Figs. 5b-c); in contrast, drainage and DIC concentrations in groundwater have an antagonist effects on DIC exports (Tabs 1, 2, 4, 5; Figs. 5b-c). As a consequence, groundwater exports the majority of DOC during the 2-3 months of high flow, but approximately the same quantity of DIC is exported during periods of high flow and periods of base flow (Tab. 5). In addition, during the study period the discharge varied by up to 100-fold (Fig. 2a); the corresponding variations in DIC and DOC concentrations and exports from the groundwater were up to 10 times (Tabs. 4, 5; Figs. 4, 5). As reported in other studies (Fiedler et al., 2006; Öquist et al., 2009), carbon export rates were mainly determined by discharge, the variations of carbon concentrations and exports being relatively small compared to the flow variation . However, for the whole sampling period, the mean weighted carbon export is almost the same both for DIC ($0.9\pm0.5$ mmol m$^{-2}$ d$^{-1}$) and DOC ($0.7\pm0.7$ mmol m$^{-2}$ d$^{-1}$) (Tab. 5), and the forest ecosystem exports in total $1.6\pm0.9$ mmol m$^{-2}$ d$^{-1}$ (equivalents to $7.0\pm3.9$ g C m$^{-2}$ yr$^{-1}$), 40% as DOC and 60% as DIC (Tab. 6). This terrestrial carbon leaching from groundwater to streams is of the same order of magnitude of carbon leaching from subsoils ($11.9\pm5.9$ g C m$^{-2}$ yr$^{-1}$) in five temperate forest plots across Europe (Kindler et al., 2011), in a temperate Japanese deciduous forests

from soils to streams (4.0 g C m$^{-2}$ yr$^{-1}$) (Shibata et al., 2005), or in European forests (9.6±3.2 g C m$^{-2}$ yr$^{-1}$) (Luyssaert et al., 2010)..

As in groundwater, DOC and DIC concentrations in first-order streams were significantly anti-correlated (Tab. 2), suggesting that dissolved carbon dynamics in streams are mostly impacted by

groundwater inputs (Kawasaki et al., 2005; Öquist et al., 2009). We could observe higher DOC concentrations in streams during early winter and high flow periods than during growing season and late summer periods (Tab. 4). Increase in DOC concentrations with discharge and high water table has been reported in the Leyre watershed (Polsenaere et al., 2013) and in many other forested catchments (Dawson et al., 2002; Striegl et al., 2005; Raymond and Saiers, 2010; Alvarez-Cobelas et al., 2012). At

our study site, during periods of high flow, first-order streams exported 0.2±0.2 mmol m$^{-2}$ d$^{-1}$ to second order streams; a flux significantly lower than DOC exports (0.7±0.7 mmol m$^{-2}$ d$^{-1}$) from groundwater to first-order streams (Tab. 5). As a consequence, during the sampling period, 70% of the groundwater DOC was either degraded or re-immobilized at the groundwater-stream interface (Tab. 5). Indeed, when groundwater DOC enters the superficial river network through drainage part of it might be rapidly

recycled by photo-oxidation (Macdonald and Minor, 2013; Moody and Worrall, 2016) or by respiration within the stream (Roberts et al., 2007; Hall Jr et al., 2016). Alternatively, DOC can be re-adsorbed on Fe- or Al-oxides particularly abundant at the river-bed oxic/anoxic interface. As a matter of fact, flocculation with Fe or Al can remove DOC from solution (Sharp et al., 2006). In contrast, DOC concentrations and exports were similar and stable in groundwater and streams during periods of base

flow (Tab. 5). This suggests that groundwater DOC behaved conservatively during low flow stages (Schiff et al., 1997), and that DOC in streams was more labile during high flow stages (Aravena et al., 2004). Indeed, in a small temperate and forested catchment in Pennsylvania (USA), McLaughlin and Kaplan, (2013) reported an increase in concentrations of labile DOC up to 27 fold during high flow stages compared to base flow conditions.

DIC concentration in streams increased during late summer period in parallel with those in groundwater (Tab. 4; Fig. 5b). Indeed, concentrations of DIC show an inverse relationship with discharge in the Leyre watershed (Polsenaere et al., 2013) and in other temperate catchments (Billett et al., 2004; Dawson and Smith, 2007), as the result of dilution with rain water and lower contribution of deep $CO_2$-enriched groundwater during high flow periods The discharge of DIC-rich groundwater supersaturated with $CO_2$, together with the oxidation of dissolved organic matter in surface waters, results in a large $CO_2$ supersaturation of rivers (Stets et al., 2009; Hotchkiss et al., 2015). The quick loss of DIC between groundwater and first-order streams is due to efficient degassing of $CO_2$ from headwaters (Fiedler et al., 2006; Venkiteswaran et al., 2014). This rapid degassing is also attested by the change in the $\delta^{13}C$ signature of the DIC (Polsenaere and Abril 2012; Venkiteswaran et al., 2014; Deirmendjian and Abril, in revisions). Furthermore, the positive correlation between degassing and export of DIC (Tab. 2) confirms that groundwater DIC is the main source of $CO_2$ degassing in superficial stream waters (Öquist et al.,2009; Hotchkiss et al., 2015). Very fast degassing was confirmed by observations in spring waters that loose up to 70% of their $CO_2$ few dozen meters downstream (Öquist et al., 2009; Deirmendjian and Abril, in revision). Venkiteswaran et al (2014) concluded that most of the stream $CO_2$ originating from groundwater drainage was degassed before typical in-stream sampling occurs. Throughout the sampling period degassing was on a yearly average approximately $0.7\pm0.5$ mmol m$^{-2}$ d$^{-1}$ (equivalent to $3.1\pm2.2$ g C m$^{-2}$ yr$^{-1}$). $CO_2$ degassing was higher during high flow periods than during periods of base flow (Tab. 5), as a consequence of higher discharge and inputs of groundwater DIC to streams (Tabs. 1, 4) and higher water turbulence. As a matter of fact, degassing depends on water velocity that induces water turbulence and thus increases the gas transfer velocity (Alin et al., 2011; Raymond et al., 2012). Overall, during the whole sampling period $CO_2$ degassing in streams represented approximately 75% of the DIC exported from groundwater and thus, a significant part of the carbon exported from forest plot rapidly returns in the atmosphere in the form of $CO_2$ through degassing.

Leaching of terrestrial carbon from the pine forest in the Leyre watershed calculated as the dissolved organic and inorganic carbon export per catchment area was $1.6\pm0.9$ mmol m$^{-2}$ d$^{-1}$ (equivalents to $7.0\pm3.9$ g C m$^{-2}$ yr$^{-1}$). Eddy covariance measurements at the Bilos plot (Tab. 3) provided a forest net uptake of atmospheric $CO_2$ of approximately $-90\pm110$ mmol m$^{-2}$ d$^{-1}$ (equivalent to $390\pm480$ g C m$^{-2}$ yr$^{-1}$). In the same way as groundwater DOC and DIC stocks represent a minor fraction of soil carbon, C leaching represents a very small (approximately 2%) fraction of forest NEE, a conclusion consistent with other studies in temperate forest ecosystems (Shibata et al., 2005; Kindler et al., 2011; Magin et al., 2017). Such weak export of carbon from forest ecosystems, at least in temperate regions, is at odds with recent studies that attempt to integrate the contribution of inland waters in the continents carbon budget (Ciais et al., 2013). Indeed, at the global scale, the quantity of terrestrial carbon necessary to account for the sum of $CO_2$ degassing from inland waters, organic carbon burial in sediments and carbon export to the ocean, represents more than 2 PgC y$^{-1}$, a number similar to the actual net land sink of atmospheric $CO_2$ (Ciais et al., 2013). Understanding why local and global carbon mass balances strongly diverge on the proportion of land NEE exported to aquatic systems appears a major challenge for the next years of research on the field.

## 5. Conclusion

The monitoring of DIC and DOC concentrations in groundwater and first-order streams in podzol-dominated catchment overlaid by pine forest, brings new insights on the nature of processes that control carbon leaching from soils, transformation in groundwater and export to surface waters and back to the atmosphere (Fig. 7). This terrestrial-aquatic-atmosphere interface is believed to behave as a hotspot in the continental carbon cycle. The permeable character of the soil at the study site enables a clear temporal decomposition of processes involving carbon in groundwaters in relation with water table depth and amplitude, forest ecosystem production and respiration. Hydrology has a strong influence on the carbon concentrations in shallow groundwater. High precipitation caused the water table to rise and saturate the topsoil, inducing a large mobilization of soil organic matter as DOC in the shallow

groundwater, a process also favored by temporary reducing conditions in the topsoil. These high water table periods are also associated with low DIC concentrations in groundwater caused by the groundwater dilution with rainwater. On the opposite way, groundwater was enriched in DIC during base flow stages, as the result of two distinct processes. First, microbial consumption of DOC occurs within the groundwater in spring and summer, the following months after the high water table periods. Second, heterotrophic conditions in the forest ecosystem during late summer favor the downward diffusion of soil $CO_2$ to shallow groundwater.

In the absence of surface runoff, the comparison of dissolved carbon concentrations between groundwater and streams, associated with drainage data, allows to understand and quantify the processes at the groundwater-stream-atmosphere interface. In the studied catchment, this method reveals a fast degassing of DIC as $CO_2$ throughout the year in first-order streams. During base flow periods, groundwater DOC was exported conservatively to streams, probably because groundwater DOC was more recalcitrant, more stabilized and more aged during this period. However, during winter and high water table, the rise of DOC concentration in groundwater observed at some site but not at others, did not fully translate to streams, some spatial heterogeneity of export in the landscape, a fast degradation and/or some re-adsorption processes in soils close to the groundwater-streams interface.

Although spatial extrapolation of quantitative information from the plot scale to first-order streams in the watershed may have generated some uncertainty, we could make a comparison of groundwater carbon export to stream with other carbon fluxes in the landscape. Representing 2% of the local forest NEE, DIC and DOC exports to surface waters do not seem a significant component of the carbon budget at our study site. More detailed work at the land-water interface is necessary in order to reconcile the contradictory findings at local and global scales on the significance of hydrological carbon export in the continental carbon budget.

## Acknowledgments

This research is part of the CNP-Leyre project funded by the Cluster of Excellence COTE at the Université de Bordeaux (ANR-10-LABX-45). We thank Luiz Carlos Cotovicz Junior, Katixa Lajaunie-Salla, Baptiste Voltz, Gwenaëlle Chaillou and Damien Buquet (EPOC Bordeaux) for their assistance in the field. We thank Pierre Anshutz (EPOC, Bordeaux), Alain Mollier and Christian Morel (ISPA INRA) for their implications on the CNP-Leyre project, and Céline Charbonnier for alkalinity titrations in the laboratory. Pierre Trichet (ISPA INRA) provided SOC data at the Bilos site.

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

|  | Precipitation (mm d⁻¹) | Evapotranspiration (mm d⁻¹) | Drainage (mm d⁻¹) | Groundwater storage (mm d⁻¹) |
|---|---|---|---|---|
| 2014 | 3.0±2.1 | 2.5±1.4 | 0.5±0.5 | -0.2±2.3 |
|  | [0.2~8.0] | [0.3~5.3] | [0.1~1.9] | [-2.9~.4.5] |
| 2015 | 1.9±1.2 | 1.7±1.0 | 0.3±0.3 | -0.5±1.9 |
|  | [0.2~4.1] | [0.3.~3.4] | [0.1~0.9] | [-3.1~2.6] |
| High flow | 4.7±2.1 | 2.4±1.0 | 1.1±0.4 | -0.2 |
|  | [2.2~8.0] | [0.9~3.6] | [0.7~1.9] | [-2.9~4.0] |
| Growing season | 1.8±0.8 | 3.0±0.9 | 0.3±0.2 | -1.9 |
|  | [0.8~2.9] | [1.6~5.3] | [0.1~0.7] | [-3.1~-0.5] |
| Late summer | 1.1±0.5 | 1.5±0.5 | 0.1±0.007 | 0.1 |
|  | [0.2~1.5] | [1.0~2.2] | [0.1~0.1] | [-1.2~0.7] |
| Early winter | 2.7±1.5 | 0.5±0.2 | 0.2±0.07 | 1.9 |
|  | [0.2~4.7] | [0.3~0.7] | [0.1~0.3] | [0.7~4.5] |

Table 1: Water budget at the Bilos plot scale for the year 2014 and 2015, as well as for high flow (Jan. 2014-Mar. 2014 and Feb. 2015-Mar. 2015), growing season (Apr. 2014-Aug. 2014 and Apr. 2015-Aug. 2015), late summer (Sep. 2014-Oct. 2014 and Sep. 2015-Oct. 2015) and early winter (Nov. 2014-Jan. 2015 and Nov. 2015-Dec. 2015). Numbers represent the mean±SD and the range (between square brackets).

| | Concentration in groundwater (mmol m$^{-3}$) | | Concentration in streams (mmol m$^{-3}$) | | Stock in groundwater (mmol m$^{-2}$) | | Export from groundwater to streams (mmol m$^{-2}$ d$^{-1}$) | | Degassing in streams (mmol m$^{-2}$ d$^{-1}$) | Hydrological parameters (mm d$^{-1}$) | | | | Water table depth (mm) | Metabolic parameters (mmol m$^{-2}$ d$^{-1}$) | | |
|---|---|---|---|---|---|---|---|---|---|---|---|---|---|---|---|---|---|
| | DIC$_{gw}$ | DOC$_{gw}$ | DIC$_{stream}$ | DOC$_{stream}$ | DIC$_{stock}$ | DOC$_{stock}$ | DIC$_{export}$ | DOC$_{export}$ | F$_{degass}$ | P | GWS | ETR | D | H | NEE | GPP | R |
| DIC$_{gw}$ | 1 | **-0.65** | **0.86** | -0.34 | **0.99** | **-0.65** | -0.44 | **-0.62** | -0.48 | -0.02 | 0.45 | -0.41 | **-0.68** | **-0.83** | 0.52 | -0.31 | -0.09 |
| DOC$_{gw}$ | | 1 | -0.41 | 0.43 | **-0.64** | **0.98** | 0.69 | **0.95** | 0.56 | 0.17 | -0.28 | 0.41 | **0.93** | **0.85** | -0.19 | -0.13 | -0.36 |
| DIC$_{stream}$ | | | 1 | **-0.55** | **0.82** | -0.42 | -0.35 | -0.34 | **-0.54** | -0.14 | 0.23 | -0.25 | -0.44 | **-0.75** | 0.43 | -0.32 | -0.16 |
| DOC$_{stream}$ | | | | 1 | -0.33 | 0.45 | 0.45 | 0.38 | **0.66** | 0.30 | 0.35 | -0.39 | 0.46 | **-0.70** | 0.15 | -0.41 | **-0.53** |
| DIC$_{stock}$ | | | | | 1 | **-0.63** | 0.37 | **-0.62** | -0.44 | -0.04 | 0.45 | -0.44 | **0.67** | **-0.79** | -0.48 | 0.28 | -0.07 |
| DOC$_{stock}$ | | | | | | 1 | **-0.82** | **0.97** | 0.67 | 0.21 | -0.23 | 0.32 | **-0.97** | **0.88** | 0.20 | -0.14 | -0.39 |
| DIC$_{export}$ | | | | | | | 1 | **0.72** | **0.86** | 0.26 | 0.01 | 0.02 | **0.83** | **0.76** | -0.18 | -0.15 | -0.39 |
| DOC$_{export}$ | | | | | | | | 1 | **0.57** | 0.24 | -0.22 | 0.28 | **0.98** | **0.81** | -0.19 | -0.17 | -0.41 |
| F$_{degass}$ | | | | | | | | | 1 | 0.45 | 0.24 | 0.17 | **0.70** | **0.78** | 0.06 | -0.35 | **-0.50** |
| P | | | | | | | | | | 1 | 0.76 | -0.30 | 0.29 | 0.23 | 0.33 | -0.44 | -0.43 |
| GWS | | | | | | | | | | | 1 | **-0.73** | -0.16 | -0.15 | **0.62** | **-0.63** | **-0.51** |
| ETR | | | | | | | | | | | | 1 | 0.22 | 0.17 | **-0.63** | **0.63** | **0.50** |
| D | | | | | | | | | | | | | 1 | **0.88** | -0.23 | -0.15 | -0.41 |
| H | | | | | | | | | | | | | | 1 | -0.27 | -0.06 | 0.31 |
| NEE | | | | | | | | | | | | | | | 1 | **-0.85** | **-0.55** |
| GPP | | | | | | | | | | | | | | | | 1 | **0.91** |
| R | | | | | | | | | | | | | | | | | 1 |

Table 2: Linear correlation (Pearson) between the studied parameters at the Bilos plot scale during the sampling period. Numbers represent the Pearson's correlation coefficient at the Bilos plot between mean carbon concentrations (mmol m$^{-3}$) in the Bilos groundwater and in the 6 first-order streams, carbon stocks (mmol m$^{-2}$ d$^{-1}$), carbon exports (mmol m$^{-2}$ d$^{-1}$), carbon degassing (mmol m$^{-2}$ d$^{-1}$) in the 6 first-order streams, hydrological parameters (in mm d$^{-1}$, which are P, GWS, ETR and D for precipitation, groundwater storage, evapotranspiration and drainage, respectively), water table depth (mm), and metabolic parameters (mmol m$^{-2}$ d$^{-1}$). Here, degassing was calculated from the DIC data of the Bilos groundwater only. Each parameter was integrated between two sampling dates (Tab. S2). Values in bold indicate correlation with p-value < 0.05, whereas underlined and bolded values indicate correlation with p-value < 0.001.

|  | GPP (mmol m$^{-2}$ d$^{-1}$) | R$_{eco}$ (mmol m$^{-2}$ d$^{-1}$) | NEE (mmol m$^{-2}$ d$^{-1}$) |
|---|---|---|---|
| 2014-2015 | 400±210 | 310±150 | -90±110 |
|  | [160~880] | [110~660] | [-340~100] |
| High flow | 300±80 | 180±50 | -120±50 |
|  | [180~420] | [105~260] | [-160~-30] |
| Growing season | 640±150 | 490±100 | -160±140 |
|  | [380~880] | [320~640] | [-330~100] |
| Late summer | 350±120 | 300±80 | -50±60 |
|  | [240~540] | [200~410] | [-160~10] |
| Early winter | 210±30 | 230±50 | 20±20 |
|  | [160~260] | [170~320] | [-10~65] |

Table 3: Metabolic parameters (GPP, R$_{eco}$ and NEE) estimated at the Bilos plot with the eddy covariance techniques. Numbers represent the mean±SD and the range (between square brackets) for the years 2014-2015, and for high flow (Jan. 2014-Mar. 2014 and Feb. 2015-Mar. 2015), growing season (Apr. 2014-Aug. 2014 and Apr. 2015-Aug. 2015), late summer (Sep. 2014-Oct. 2014 and Sep. 2015-Oct. 2015) and early winter (Nov. 2014-Jan. 2015 and Nov. 2015-Dec. 2015) periods. Positive NEE indicates an upward flux whereas a negative NEE indicates a downward flux, GPP is positive or zero and R$_{eco}$ is positive. NEE = R$_{eco.}$- GPP.

| | DOC (mmol m$^{-3}$) | | | | DIC (mmol m$^{-3}$) | | | |
|---|---|---|---|---|---|---|---|---|
| | Piezometer Bilos | Piezometer 2 | Piezometer 3 | Streams | Piezometer Bilos | Piezometer 2 | Piezometer 3 | Streams |
| High flow | 3,500±200 [3,200~3,700] N=3 | 280 N=1 | 1,500 N=1 | 490±10 [460~510] N=15 | 1,160±470 [570~1,700] N=3 | 1,380 N=1 | 1,510 N=1 | 280±40 [220~310] N=15 |
| Growing season | 750±440 [320~950] N=7 | 380±40 [300~400] N=5 | 880±400 [550~830] N=4 | 360±100 [200~540] N=41 | 2,570±240 [2,350~3,030] N=7 | 1,450±380 [1,000~2,100] N=5 | 2,030±220 [1,650~2,160] N=4 | 330±120 [210~550] N=41 |
| Late summer | 540±60 [480~600] N=2 | 420±80 [340~500] N=2 | N=0 | 370±30 [340~400] N=4 | 5,240±140 [5,100~5,400] N=2 | 3,900±100 [3,800~4,000] N=2 | N=0 | 1,030±240 [790~1,270] N=4 |
| Early winter | 640±50 [580~670] N=3 | 470±110 [350~620] N=3 | 760 N=1 | 510±30 [480~550] N=17 | 2,600±980 [1,850~4,000] N=3 | 2,370±1,500 [940~4,500] N=3 | 2,040 N=1 | 300±90 [240~430] N=17 |

Table 4: Carbon concentrations in the sampled groundwater and in the sampled first-order streams during the sampling period (Jan. 2014-Jul. 2015) for high flow (Jan. 2014-Mar. 2014 and Feb. 2015-Mar. 2015), growing season (Apr. 2014-Aug. 2014 and Apr. 2015-Aug. 2015), late summer (Sep. 2014-Oct. 2014 and Sep. 2015-Oct. 2015) and early winter (Nov. 2014-Jan. 2015 and Nov. 2015-Dec. 2015) periods. Numbers represent the mean±SD, the range (between square brackets) and the number (N) of samples for each hydrological period.

| | DOC$_{export}$ mmol m$^{-2}$ d$^{-1}$ | | | | DIC$_{export}$ mmol m$^{-2}$ d$^{-1}$ | | | | Degassing mmol m$^{-2}$ d$^{-1}$ |
|---|---|---|---|---|---|---|---|---|---|
| | Bilos piezometer | Piezometer 2 | Piezometer 3 | Streams[b] | Bilos piezometer | Piezometer 2 | Piezometer 3 | Streams[b] | Streams |
| High flow | 3.4±1.1 | 0.4±0.02 | 1.5±0.2 | 0.6±0.1 | 1.8±0.4 | 1.4±0.2 | 1.8±0.1 | 0.3±0.1 | 1.4±0.2 |
| | [2.3~4.9] | [0.3~0.4] | [1.2~1.7] | [0.5~0.7] | [1.3~2.2] | [1.3~1.6] | [1.7~1.9] | [0.3~0.4] | [0.8~1.9] |
| Growing season | 0.4±0.4 | 0.05±0.02 | 0.2±0.2 | 0.1±0.1 | 0.7±0.3 | 0.3±0.1 | 0.6±0.1 | 0.1±0.03 | 0.5±0.2 |
| | [0.1~1.2] | [0.1~0.2] | [0.1~0.4] | [0.05~0.3] | [0.4~1.3] | [0.3~0.5] | [0.4~0.7] | [0.05~0.2] | [0.3~1.3] |
| Late summer | 0.1±0.01 | 0.1±0.04 | | 0.1±0.01 | 0.6±0.03 | 0.4±0.05 | | 0.1±0.01 | 0.4±0.1 |
| | [0.1~0.1] | [0.1~0.1] | | [0.05~0.1] | [0.6~0.7] | [0.4~0.5] | | [0.1~0.1] | [0.4~0.6] |
| Early winter | 0.1±0.02 | 0.1±0.03 | 0.2 | 0.1±0.02 | 0.7±0.1 | 0.6±0.2 | 0.6 | 0.1±0.02 | 0.5±0.1 |
| | [0.1~0.2] | [0.1~0.1] | | [0.1~0.1] | [0.5~0.8] | [0.4~0.8] | | [0.1~0.1] | [0.5~0.6] |
| 2014-2015 | 0.9±1.4 | 0.1±0.1 | 0.6±0.5 | 0.2±0.2 | 0.9±0.5 | 0.6±0.4 | 1.0±0.6 | 0.2±0.1 | 0.6±0.3 |
| | [0.1~4.9] | [0.05~0.4] | [0.1~1.7] | [0.05~0.7] | [0.4~2.2] | [0.3~1.6] | [0.4~1.9] | [0.05~0.4] | [0.2~1.3] |
| Entire watershed (2014-2015) | 0.7±0.7[a] | | | 0.2±0.2 | 0.9±0.5[a] | | | 0.2±0.1 | 0.7±0.5 |

Table 5: Export of DIC and DOC from the sampled groundwater to first-order streams, as well as degassing in first-order streams; for the sampling period and for high flow (Jan. 2014-Mar. 2014 and Feb. 2015-Mar. 2015), growing season (Apr. 2014-Aug. 2014 and Apr. 2015-Aug. 2015), late summer (Sep. 2014-Oct. 2014 and Sep. 2015-Oct. 2015) and early winter (Nov. 2014-Jan. 2015 and Nov. 2015-Dec. 2015) periods. Numbers represent the mean±SD whereas numbers between square brackets represent the range. Here, degassing was calculated with the DIC data from the 3 sampled groundwaters. [a] represents the mean carbon export weighted by surface assuming that the Bilos piezometer is representative of the wet Landes, that Piezometer 2 is representative of the dry Landes and that Piezometer 3 is

055

representative of the mesophyllous Landes and using the relative surface area of each type of Landes. [b] represents carbon

exports from first to second order streams and it have been calculated from the drainage of first-order streams (mm d$^{-1}$)

and the mean concentrations of DOC and DIC in first-order streams (mmol m$^{-3}$).

Figure captions

Figure 1: Map of the Leyre watershed with topography showing the location of the gauging stations (the Grande Leyre, the Petite Leyre, the Grand Arriou and the Bourron rivers), the Bilos site, as well as the locations of the other sampled piezometers and first-order streams. Rain gauge and Eddy tower are located at the Bilos plot. White circles indicate the first-order streams where additional discharge measurements have been made in Apr. 2014 and Feb. 2015.

Figure 2: Seasonal variations of hydrological parameters in the Leyre watershed. (a) Discharge of the Grande Leyre, the Petite Leyre, the Grand Arriou and the Bourron rivers associated with water table at the Bilos site; (b) Metabolic parameters (NEE, GPP, $R_{eco}$) estimated at the Bilos site; (c) Monthly precipitation, evapotranspiration and groundwater storage at the Bilos site as well as the drainage of first-order streams. Inputs of water (precipitation and positive groundwater storage) in the studied ecosystem are represented on a positive scale whereas outputs of water (drainage, evapotranspiration and negative groundwater storage) are represented on a negative scale. HF, GS, LS and EW represent respectively high flow (Jan. 2014-Mar. 2014 and Feb. 2015-Mar. 2015), growing season (Apr. 2014-Aug. 2014 and Apr. 2015-Aug. 2015), late summer (Sep. 2014-Oct. 2014 and Sep. 2015-Oct. 2015) and early winter (Nov. 2014-Jan. 2015 and Nov. 2015-Dec. 2015) periods.

Figure 3: Monthly water mass balance at the Bilos site for the 2014-2015 period. Pearson coefficient R = 0.85, p-value < 0.001. Blue points represent months where GWS was extremely negative in Mar. 2014, Apr. 2014, Mar. 2015, Apr. 2015, Jun. 2015 and Jul. 2015) (Fig. 2c). These blue points are further

away from the 1:1 Line than the other months (represented in black). The drainage of the Leyre River is delayed compared to the drainage of the Bilos plot. Thus, when the loss of groundwater is extremely high (negative GWS), estimated drainage do not correspond exactly to the measured groundwater storage.

Figure 4: The Concentrations of DIC and DOC in the three sampled groundwater as a function of water table depth.

Figure 5: (a) Discharge of the Grande Leyre, the Petite Leyre, the Grand Arriou and the Bourron rivers associated with water table at the Bilos site. Temporal variations throughout the sampling period of (b) the DIC concentrations in the sampled piezometers and in the sampled first-order streams (medium dashed line; errors bars represent standard deviation of the six first-order streams) and of (c) the DOC concentrations in the sampled piezometers and in the sampled first-order streams (medium dashed line; errors bars represent standard deviation of the six first-order streams). HF, GS, LS and EW represent respectively high flow (Jan. 2014-Mar. 2014 and Feb. 2015-Mar. 2015), growing season (Apr. 2014-Aug. 2014 and Apr. 2015-Aug. 2015), late summer (Sep. 2014-Oct. 2014 and Sep. 2015-Oct. 2015) and early winter (Nov. 2014-Jan. 2015 and Nov. 2015-Dec. 2015) periods.

Figure 6: (a) The mean DIC and DOC stocks between two sampling dates in Bilos groundwater. HF, GS, LS and EW represent respectively high flow (Jan. 2014-Mar. 2014 and Feb. 2015-Mar. 2015), growing season (Apr. 2014-Aug. 2014 and Apr. 2015-Aug. 2015), late summer (Sep. 2014-Oct. 2014 and Sep. 2015-Oct. 2015) and early winter (Nov. 2014-Jan. 2015 and Nov. 2015-Dec. 2015) periods.

Figure 7: Conceptual model at the vegetation-soil-groundwater-stream interface in sandy ecosystems having shallow groundwater. OH, WT and D are the organic horizon of the soil, the water table and the drainage, respectively. Hydro-biogeochemical processes are represented in dashed arrows. Carbon exports are represented in full arrows; the thickness of the arrow indicates the magnitude of flux. High flow periods are in Jan. 2014-Mar. 2014 and Feb. 2015-Mar. 2015, growing season in Apr. 2014-Aug. 2014 and Apr. 2015-Aug. 2015, late summer in Sep. 2014-Oct. 2014 and Sep. 2015-Oct. 2015 and early winter in Nov. 2014-Jan. 2015 and Nov. 2015-Dec. 2015.

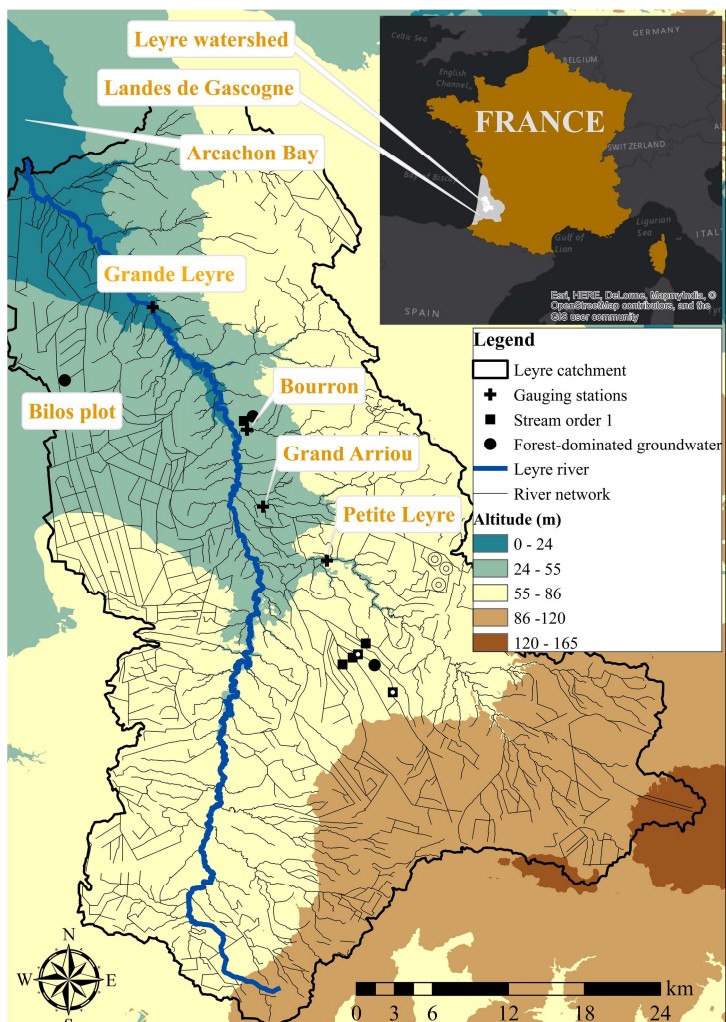

Figure 1

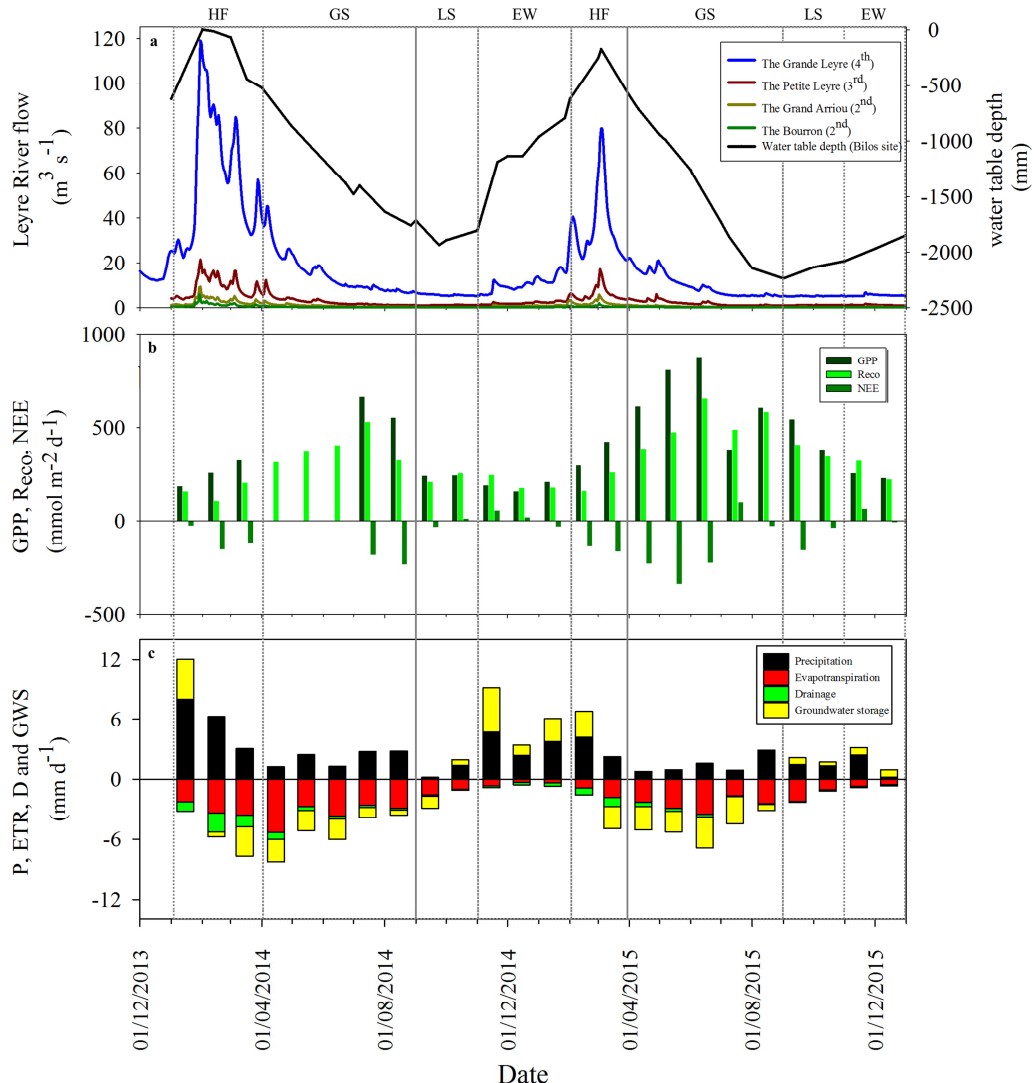

Figure 2

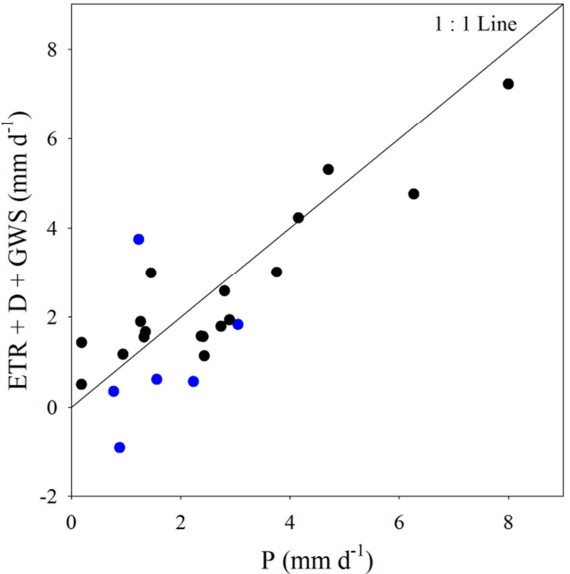

Figure 3

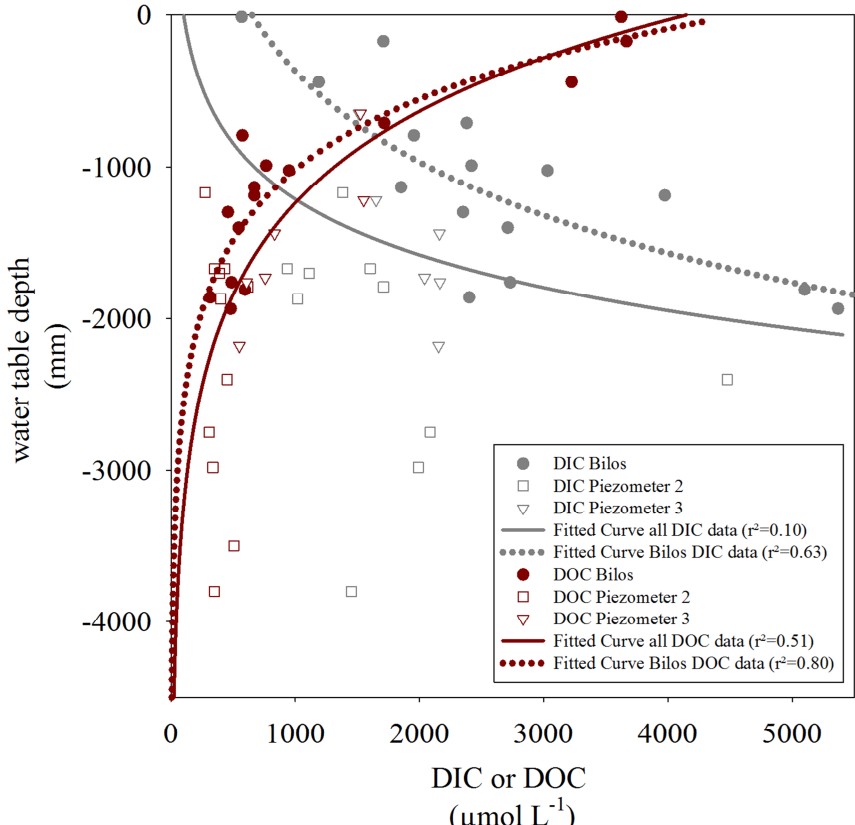

Figure 4

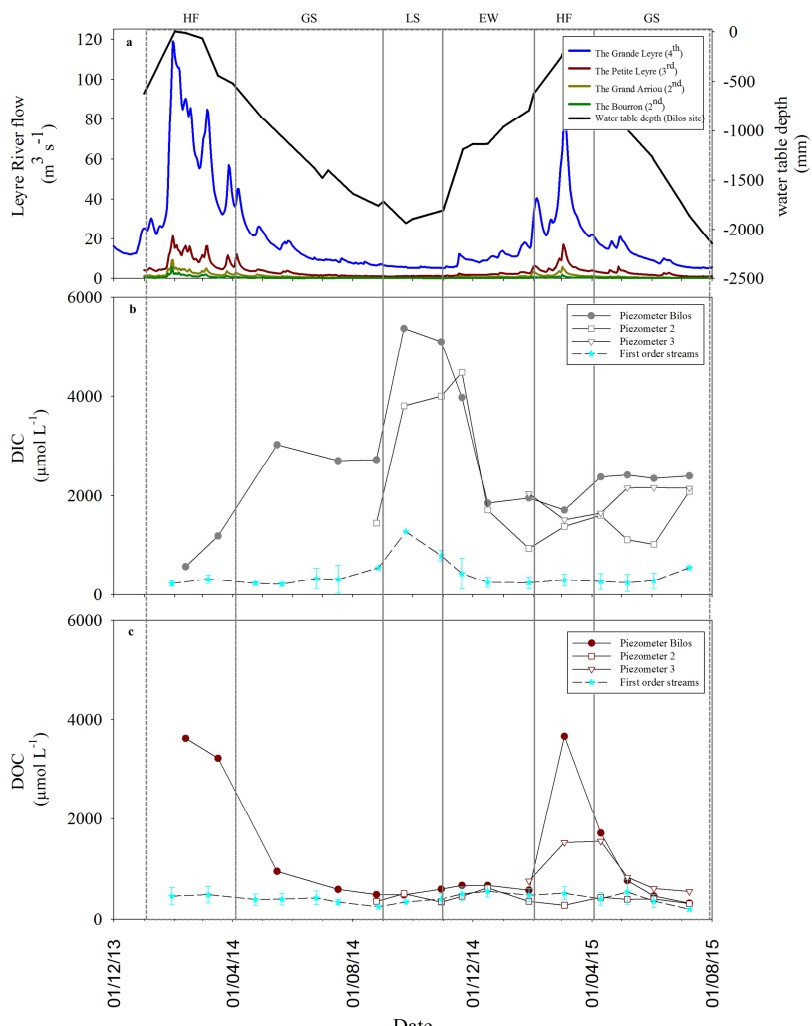

Figure 5

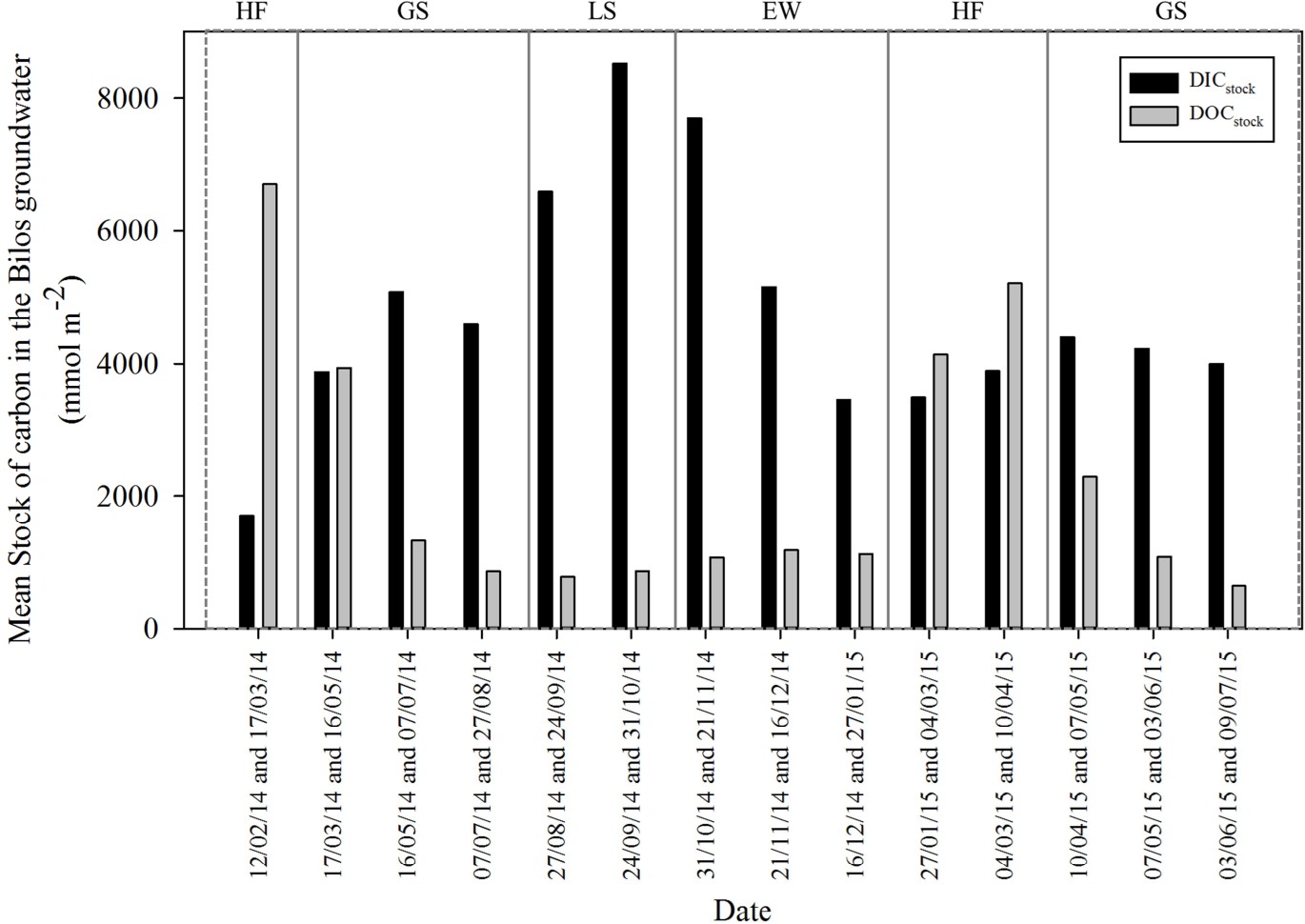

Figure 6

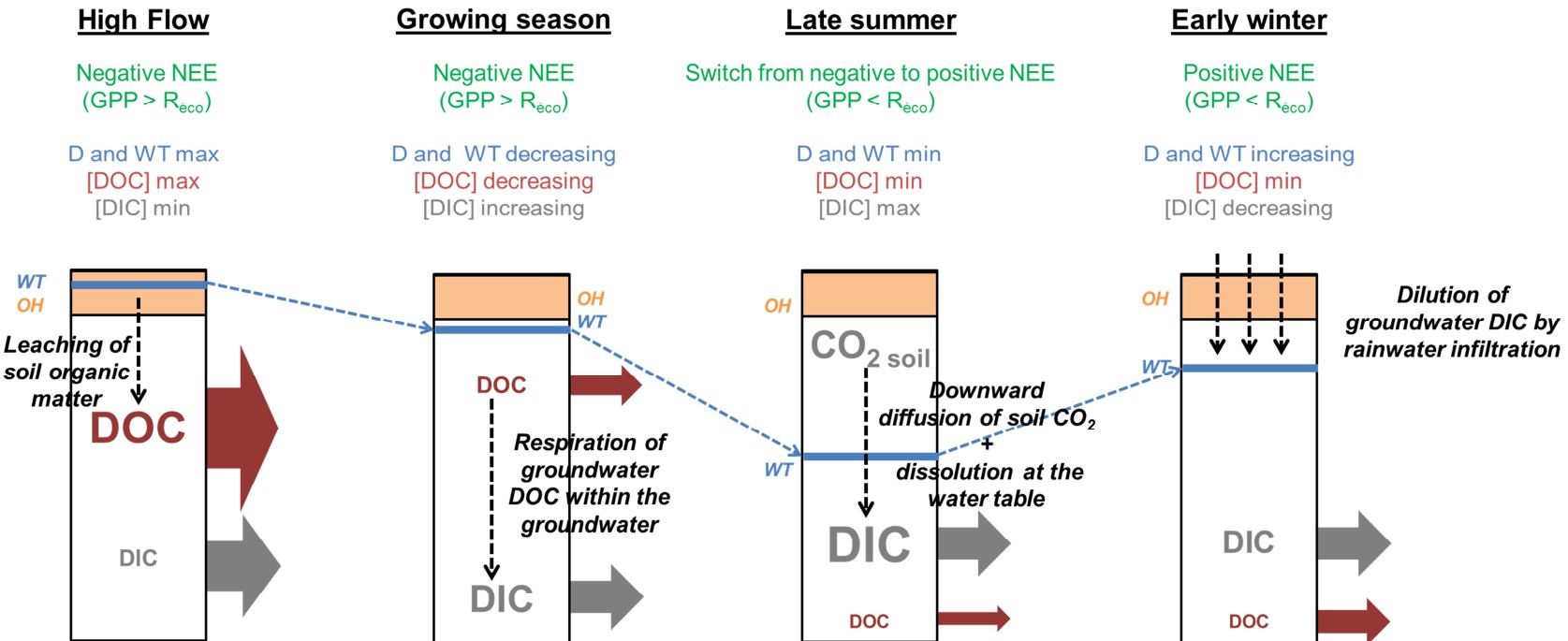

Figure 7

.125

Supplementary Material

| Period | Date | Groundwater | | | Surface water |
|---|---|---|---|---|---|
| | | Piezometer Bilos | Piezometer 2 | Piezometer 3 | First-order streams |
| HF | 29/01/2014 | | | | X |
| HF | 12/02/2014 | X | | | |
| HF | 07/03/2014 | | | | X |
| HF | 17/03/2014 | X | | | |
| GS | 24/04/2014 | | | | X |
| GS | 16/05/2014 | X | | | |
| GS | 21/05/2014 | | | | X |
| GS | 25/06/2014 | | | | X |
| GS | 17/072014 | X | | | X |
| GS | 27/08/2014 | X | X | | X |
| LS | 24/09/2014 | X | X | | X |
| LS | 31/10/2014 | X | X | | X |
| EW | 21/11/2014 | X | X | | X |
| EW | 16/12/2014 | X | X | | X |
| EW | 27/01/2015 | X | X | X | X |
| HF | 04/03/2015 | X | X | X | X |
| GS | 10/04/2015 | X | X | X | X |
| GS | 07/05/2015 | X | X | X | X |
| GS | 03/06/2015 | X | X | X | X |
| GS | 09/07/2015 | X | X | X | X |

Table S1: Sampling dates of groundwater and surface waters. X correspond to a sampling of $pCO_2$, total alkalinity and

DOC. HF, GS, LS and EW represent respectively high flow (Jan. 2014-Mar. 2014 and Feb. 2015-Mar. 2015), growing

130 season (Apr. 2014-Aug. 2014 and Apr. 2015-Aug. 2015), late summer (Sep. 2014-Oct. 2014 and Sep. 2015-Oct. 2015)

and early winter (Nov. 2014-Jan. 2015 and Nov. 2015-Dec. 2015) periods.

| Period | Date | DIC$_{stock}$ and DOC$_{stock}$ | DIC$_{export}$ and DOC$_{export}$ | | | F$_{degass}$ |
|---|---|---|---|---|---|---|
| | | Piezometer Bilos | Piezometer Bilos | Piezometer 2 | Piezometer 3 | Streams |
| HF | 12/02/14 and 17/03/14 | X | X | | | X[a] |
| GS | 17/03/14 and 16/05/14 | X | X | | | X[b] |
| GS | 16/05/14 and 17/07/14 | X | X | | | X[c] |
| GS | 17/07/14 and 27/08/14 | X | X | | | X |
| LS | 27/08/14 and 24/09/14 | X | X | X | | X |
| LS | 24/09/14 and 31/10/14 | X | X | X | | X |
| EW | 31/10/14 and 21/11/14 | X | X | X | | X |
| EW | 21/11/14 and 16/12/14 | X | X | X | | X |
| EW | 16/12/14 and 27/01/15 | X | X | X | | X |
| HF | 27/01/15 and 04/03/15 | X | X | X | X | X |
| HF | 04/03/15 and 10/04/15 | X | X | X | X | X |
| GS | 10/04/15 and 07/05/15 | X | X | X | X | X |
| GS | 07/05/15 and 03/06/15 | X | X | X | X | X |
| GS | 03/06/15 and 09/07/15 | X | X | X | X | X |

Table S2: Periods of calculation for carbon stocks, carbon exports and carbon degassing. X corresponds to a calculation.

[a, b, c] for these periods the day of sampling of surface waters do not correspond exactly to the day of sampling of groundwater (Tab. S1). Carbon stocks in groundwater can be calculated only for Bilos plot since we do not have data about the total height of the permeable layer in the other plots. HF, GS, LS and EW represent respectively high flow (Jan. 2014-Mar. 2014 and Feb. 2015-Mar. 2015), growing season (Apr. 2014-Aug. 2014 and Apr. 2015-Aug. 2015), late summer (Sep. 2014-Oct. 2014 and Sep. 2015-Oct. 2015) and early winter (Nov. 2014-Jan. 2015 and Nov. 2015-Dec. 2015) periods.