# Peer review of "Hydro-ecological controls on dissolved carbon dynamics in groundwater and export to streams in a temperate pine forest"

_Biogeosciences, 2017_

## Referee Comment (RC1) · Anonymous Referee #1 · 7 May 2017

Major comments- The study address an important question and could be a significant contribution to the field of carbon biogeochemistry. Methods and sampling design are appropriate but the paper suffers from poor organization and presentation. Furthermore, there are several other critical issues that authors need to address before manuscript can be recommended for publication. Firstly, authors broadly discussed some knowledge gap in the Introduction section but no clarity on why this study was needed at the first place. What is necessary is a more thorough review of similar studies in forested catchments and clearly working out the current knowledge gap that authors attempted to address. Secondly, the Result section is too detailed, unguided, and

extremely difficult to follow. The reader has no idea what is coming up next and what are the take away messages from each paragraph. Thirdly, the Discussion section is detailed, but somewhat unguided and does not show : a) how the study advances our general understanding of groundwater's role in mediating carbon exports from forested catchments, and b) how this study builds on the past work. Fourthly, massive overuse of the term "control". Control is a pretty strong word in the Catchment Science and I am not sure if the current sampling design (i.e., 3 wells, 1 rain gauge) is robust enough to reveal hydrological controls of carbon exports for a 2100 km2 catchment. Lastly, there are several typos and grammatical errors throughout the draft, and I have only highlighted a few of them here.

Minor comments: Abstract: line20-21: When you say drainage here, did you mean to say- subsurface flow? Unclear? Line 21: Instrumented or implemented? Please revise this sentence. Line 29-30: Grammatical issue, please revise the sentence. Can you please add a sentence on the major implication of the study?

Introduction Line 59-60: Please add references? Line 62: How so, please explain? No references, is this the first study to address this question?

I can understand the overall aim of the study, but it would be helpful if you could clearly layout research questions or even hypotheses.

Methods Line 76: Several places you have used "per mill" symbol. Line 92: You are focusing on shallow groundwater, may be worth mentioning once for clarity? Line 97: Some abbreviations are not in the table 1; SOC? Line 110: Sampled for what ? Line 166: How did you interpolate groundwater stages between two dates? Line 196: How did you estimate D, please explain. I am extremely confused about how you took the catchment scale response from a stream gauge, and after multiplying it by a factor of 2 assumed it is as discharge for the bilos site? Line 214: With monthly sampling frequency, you will miss a lot of variation in DOC/DIC storage, especially during storms, no? Line 233, Equation 7 will be followed by the equation 6? Line 246 In equation 8,

how did you estimate Qmean, unclear?

Results and Discussion- Too many ideas in a single paragraph, and the main message is lost in the details. It is left to the reader to figure out the take home message. I would recommend re-writing results and discussion sections for better presentation of the key ideas. Line 354: Given the study design, this sentence is a major overstatement. Line 460: Soil DOC in shallow layers cannot be mobilized by Precipitation? Lack of correlation between Soil DOC and P is not surprising as DOC exports from soil horizon is highly non-linear process. Line 490: Sentence unclear

I would recommend adding an Implication section at the end of the discussion section.

Conclusion: I am not sure you have summarized your work clearly. Line 556: Thus?

Tables and Figures Table 5. It is good to provide conceptual table and later figure 7, but please remember you can only interpret so much from your limited study design. Fig1. If possible, please show locations of the rain gauge and flux tower (or location of Eddy covariance measurements) on the map. Fig4: There is no gray side bar in the figure. Please revise caption or add the bar to the figure. How did you estimate error bars, $\pm 1$ standard deviation? Fig7. The texts within the boxes are bit small, if possible please increase their font sizes.

---

## Referee Comment (RC2) · Anonymous Referee #2 · 25 May 2017

General comment

Interesting, significant work, but presentation is poor and unacceptable. One of the reason for this is insufficient reviews of similar studies. Although the studies in which DOC/DIC exports are compared with NEE measured by eddy covariance method in the same site may be limited, there are many studies discussed about DOC/DIC exports and their mechanisms.

The result section is hard to catch up. It should be rearranged to show more clearly to guide the readers for your discussion. The discussion section is also need to rearrange.

[Figure]

I suppose that the contents in this is version of manuscript may be true, but may be incorrect at this moment.

Refer the specific comments.

Specific comments L.59 '... generally'. Here, the authors must refer some studies. Moreover, clarify how different this study from these studies?? L.74. The Leyre watershed is very large, but the piezometers are only three. Consider and clarify whether or how the data shown here represent whole the watershed. In other words, do the results in Bilos site (0.6km2) represent the whole Leyre watershed? L. 80 & 90 XIXth. Change to Arabic. L. 91 'Consequently...' References about hydrology in the Leyre watershed must be sited around this sentence. If the authors does not cite any studies here, the descriptions are suspicious. L.103. Although soil preparation, fertilization and seeding was done in 2005, the referred paper was published in 2003. Why? Is this correct? L.180. What is the 'CO2 SYS'? No information. L.258-. Result. Too complicated. I highly recommend the authors to reconsider what are your main points. L. 376. 'groundwater uptake'. This term is suspicious. Generally, many plants including pine trees use the soil water within unsaturated zone, and cannot survive if the root immersed in groundwater, saturated zone. L. 378- and Table 2. The discussions are generally based on the Pearson's correlation coefficient. However, the real hydrological and biogeochemical phenomena cannot be sufficiently described by this kind of simple value; for example, how do you explain the time lag among precipitation, response of groundwater table, and drainage? The relationship between the groundwater storage and drainage in your site (L.381) is unclear in your consideration, but I suppose that it means the nonlinearity between these parameters, and it cannot be shown by the Pearson's correlation coefficient. The authors also mentioned about this at L.382, but why this occurs in the flat topography of the watershed? Any references? I think this occurs not only in flat watershed, but also in steep watershed. Again, many discussions are not supported by reliable previous studies, previous knowledge. L.388-395. The discussion about (annual?) soil water storage is not clear for me. The authors

mentioned that it was larger in 2015 (126mm) than in 2014 (71mm). Is this correct? The annual rainfall was much smaller in 2015, and the soil water storage will be smaller as well as the groundwater storage. If the authors can show the data of soil water content and/or hydraulic head, we can get more reliable information. L. 402. Insert period after 'bicarbonates'. L. 463. Discussion of this paragraph is strange. (L. 470) 'Thus, when the forest ecosystem is a source of CO2 for the atmosphere, it is also a source of CO2 for the underlying groundwater.' ... Even under the drought condition or even when the ecosystem act as a sink of CO2, below ground part of the vegetation 'only' act as theCO2 source; respiration by root always occur. Moreover, degradation of organic carbon, including DOC always occur. It's a respiration by microorganisms. L.482 and other parts. Mean residence time, define by S/flux in this manuscript, should be reword as 'apparent turnover time'. In addition, it was calculated between two sampling days. This method cannot consider the time lag between the storage and output flux, as same as mentioned above. The Pearson's correlation coefficient cannot explain this. L. 503. The comparison with peat systems in less meaningful. Should be compared to similar ecosystem with your site. Moreover, I agree with the contents of the four referred papers (L.503-507), but how related them with your results? As I mentioned above, your discussion does not consider about the time lag, or times for decomposition and transport. The referred studies essentially mentioned about this. After these processes, SOC will move as DOC or DIC. L. 508. What's the meaning of this sentence here? L. 511. 'reported' ... Need appropriate citation. 'elsewhere'... ??? Where? Too irresponsible!! L. 512. As I mentioned at L. 74, I wonder how the authors can show that the observed groundwater is representative of whole the watershed. I think the authors cannot discuss about the spatial variation of data. This is one of the weak point of this paper. L.513-. 'Also, ...'. I agree with the first part of this sentence (but need some references about this phenomena), but cannot agree with the second part. Why the absence of correlation support the phenomena? The correlation is just a correlation; it cannot attest the phenomena. Comparison of concentration is different from the correlation. L. 517. (Righi and Wilbert, 1984) This reference is too old.
The studies about DOC quality is making steady progress. I think the authors can find more up-to-date studies to support your findings. L. 521. There are no data, analysis, and discussion about the relationship between the topography and DOC concentration. How and why the difference of DOC concentration between three piezometers was occurred by the effect of topography? L. 525. The authors have mentioned about the possibility of photodegradation (or photo oxidation) of groundwater DOC at L. 514. However, the authors also mentioned as 'DOC was not labile, and not degraded in the superficial river network' during base flow period. These two comments make me confuse. Generally, the base flow condition occurs in fine (or no rain) days. Why the photodegradation does not occur under base flow condition? Which comment is true for you (and for us)?

---

## Author Comment (AC1) · 12 Jul 2017

Dear Reviewer,

First of all we would like to thank you for the time and effort you spent on reviewing our submitted MS. We very much appreciate your comments that clearly identify some parts of the submitted MS that need more effort, clarification and re-organization. We will try to address all the questions and comments you raised and we are convinced that the revised MS will be improved significantly.

Best regards,

Loris Deirmendjian & Co-authors

**Comment#1:** *"Major comments- The study address an important question and could be a significant contribution to the field of carbon biogeochemistry. Methods and sampling design are appropriate but the paper suffers from poor organization and presentation. Furthermore, there are several other critical issues that authors need to address before manuscript can be recommended for publication."*

**Reply#1:** We thank the Referee 1 for his/her constructive comment concerning the organization and presentation of our submitted MS. This general evaluation is very consistent with that expressed by the Referee 2. We clearly realized that the submitted MS suffered from poor organization and presentation and needed a complete re-organization and a more thorough review of similar studies in forested catchments. Consequently, we will put important effort in improving our revised MS to address the questions of presentation, description of the state of the art, and discussion in the light of previous works. We will modify the Introduction section and profoundly rework the text of the Results and Discussion sections of our revised MS that will include 2 new figures (Fig. 7 and 8 in the revised MS) and 5 revised figures (Fig. 1, 2, 4, 5, 6 in the revised MS) (see below). The figure 3 of the submitted MS did not change (see below). The figure 6 and 7 of the submitted MS were removed (see below).

**Comment#2:***"Firstly, authors broadly discussed some knowledge gap in the Introduction section but no clarity on why this study was needed at the first place. What is necessary is a more thorough review of similar studies in forested catchments and clearly working out the current knowledge gap that authors attempted to address"*

**Reply#2**: As suggested by the Referee 1, we will rewrite the Introduction section, first mentioning the significant role played by forests in the global carbon cycle, acting as a net annual carbon sink. Second, we will introduce the significance of C hydrological export from terrestrial to aquatic system in the continental C budget. Then, we will highlight the fact that this export term is generally calculated as the sum of other fluxes occurring in river systems: $CO_2$ outgassing + OC burial in sediments + export downstream, and rarely from direct measurement at the land-water interface. We

have performed a thorough review of similar study in forested and other type of catchments, which revealed that, the few available works that estimate lateral export of carbon are based on observations (1) in soil water (unsaturated zone) combined with soil water models (Öquist et al., 2009; Kindler et al., 2011; Leith et al., 2015), or (2) based on differences in the flux between upper and lower stream reaches (Shibata et al., 2001, 2005), or (3) based on observations in stream water combined with stream discharge (Dawson et al., 2002; Billett et al., 2004; Dinsmore et al., 2010; Olefeldt et al., 2013). We found few works that reported DIC/DOC concentrations and dynamics in groundwater (saturated zone). Kawasaki et al ( 2005) reported DIC/DOC dynamics in groundwater but this paper is out of the scope of our watershed with sandy podzols because it focused on a watershed with granite rocks overlied by cambisols. Venkiteswaran et al (2014) compared DIC concentrations in groundwater, streams and lakes, but this paper focused on degassing in streams based on isotopic model. Artinger et al (2000), Baker et al (2000) and Shen et al (2015) reported DOC dynamics and concentrations in groundwater but these works focus on the characterization and/or metabolism of DOC and not on carbon export to streams. Consequently, these studies do not allow a complete understanding of the link between C hydrological export and the physical and biological processes occurring in soils and groundwater. In addition, the approaches based with stream discharge may miss part of the DIC export flux that occurs as excess $CO_2$ that rapidly degas upstream of the sampling points in streams.

**Changes in the revised MS#2**: At the end of the Introduction section of the revised MS, we will define clearly the objectives of our study as: (i) to investigate (among the first time for DIC) the temporal dynamics of dissolved carbon species in groundwater in relation with hydrological processes in the soil (in particular migration of the water table, groundwater mass balance and apparent turnover time) and metabolic activity of the forest ecosystem (coupling dissolved carbon studies to ecosystem atmospheric carbon exchange is also original); (ii) to compare spatio-temporal variations of carbon concentrations in groundwater and first order streams in order to study the fate of dissolved C species at the groundwater-stream interface; (iii) to quantify directly the lateral drainage of DIC and DOC from groundwater to first order streams in the case of the sandy catchment of the Leyre River where no surface runoff occurs, based on concentrations in groundwater and discharge data; (iv) to compare this lateral drainage of terrestrial C with net $CO_2$ exchange of the forest ecosystem and degassing in first order streams.

The introduction of the revised MS will be clearer and more focused than that in the first version of the MS.

**Comment#3**: *"Secondly, the Result section is too detailed, unguided, and extremely difficult to follow. The reader has no idea what is coming up next and what are the take away messages from each paragraph."*

**Reply#3:** We agree with the need of better organizing the Results section, a comment also shared by Referee 2.

**Changes in the revised MS#3:** In the revised MS, we will split the Results section in five paragraphs: (i) water mass balance; (ii) carbon net ecosystem exchange; (iii) dissolved carbon in groundwater; (iv) dissolved carbon in first order streams; (v) hydrological carbon fluxes.

Firstly, in the 3.1 section, based on hydrological and eco-physiological criteria we will distinguish 4 different hydrological periods within the sampling period (Jan. 2014-Jul. 2015): HF, GS, LS and EW periods, respectively for high flow, growing season, late summer and early winter periods (see revised Fig. 2). Then, results will be described according to these 4 hydrological periods in the revised text as well as in the different revised figures, which we believe, will lead to greater clarity and understanding.

Secondly, we will modify Fig.4 (of the submitted MS) by separating the distribution of DIC and DOC concentrations in groundwater as a function of water table depth, from the time-course of concentrations (revised Fig. 4 and 5) (see below). The new figure 4 is central for the description of dissolved carbon mobilization from soil and transfer to groundwater. We will also add new Figures 7 and 8, which describe a conceptual model of carbon transfer at the groundwater-stream interface based on our observations. These two new figures allowed us to better discuss the seasonality of carbon fluxes in interaction with hydrological and biogeochemical processes without excessive interpretations.

Thirdly, as suggested by both referees, results of the Pearson's correlation will be much less emphasized in the revised version of the MS, as they appeared too simplistic to fully explain biogeochemical and hydrological processes controlling dissolved C concentrations and fluxes. We also removed the figure 7 of the submitted MS, which was based on Pearson's parameters.

**Comment#4:** *"Thirdly, the Discussion section is detailed, but somewhat unguided and does not show: a) how the study advances our general understanding of groundwater's role in mediating carbon exports from forested catchments, and b) how this study builds on the past work."*

**Reply#4:** We agree with the need of better organizing the Discussion section, a comment also shared by Referee 2. However, we would like to mention first that "past work" is quiet embryonic, so it is not easy to build on that. Indeed, very few studies compare groundwater and streams (Kawasaki et al., 2005; Venkiteswaran et al., 2014). We found few papers that report DOC dynamics in groundwater of forest ecosystem (Artinger et al., 2000; Baker et al., 2000; Shen et al., 2015) but those references are out of the scope of our study because they focused on the characterization and/or metabolism of groundwater DOC and not on carbon export to streams.

In addition, our study cannot exactly helps our "understanding of groundwater's role in mediating carbon export from forested catchments" because in our particular case, we have selected on purpose a watershed with sandy and permeable soils where no surface runoff occurs. As a consequence, at our study site, groundwater mediates all the C export to the watershed. However, our study helps understanding the dynamics of C export through groundwater drainage.

**Changes in the revised MS#4:** In the revised MS, we have split the Discussion section in three paragraphs: (i) water mass balance; (ii) soil carbon leaching to groundwater; (iii) carbon transfer at the groundwater-stream interface.

In addition, we will clarify the Discussion section in the revised MS and add more literature and discussion about DIC/DOC export from land to surface waters. However, we will also stress the lack of studies of carbon in groundwater, the knowledge gap at the soil-groundwater interface, and how our study helps in filling these gaps.

We will also modify significantly our figures by splitting the submitted Fig. 4 (panel a and panels b&c) into two separate figures (revised Fig.4 and 5), insisting in the description of the new figure 4 in the text. Indeed, the opposite evolution of DIC and DOC with water table is a first key result of our work that needs to be more highlighted. We will also show additional discharge data in first, second, and third order streams that will allow a much better description of the hydrological model used in the watershed.

**Comment#5:** *"Fourthly, massive overuse of the term "control". Control is a pretty strong word in the Catchment Science and I am not sure if the current sampling design (i.e., 3 wells, 1 rain gauge) is robust enough to reveal hydrological controls of carbon exports for a 2100 km2 catchment."*

**Reply#5:** Indeed, this term was overused in the submitted MS.

**Changes in the revised MS#5**: In the revised MS, we will change wording using also the terms "relation" or "impact", and using other arguments than the Pearson correlation coefficient.

**Comment#6:** *" Abstract: Iine20-21: When you say drainage here, did you mean to say- subsurface flow? Unclear?"*

**Reply#6:** In our study, drainage referred to shallow groundwater drainage.

**Changes in the revised MS#6:** In the revised MS, we will clarify the term drainage in the Materials and Methods section.

**Comment#7:** *"Line 21: Instrumented or implemented? Please revise this sentence."*

**Reply#7**: Instrumented.

**Changes in the revised MS#7:** We will rewrite the sentence as follows: "The studied watershed was also instrumented for continuous measurements of shallow groundwater table depth, precipitation, evapotranspiration, river discharge, and net ecosystem exchanges of sensible and latent heat fluxes as well as $CO_2$ fluxes.

**Comment#8:** *"Line 29-30: Grammatical issue, please revise the sentence"*.

**Reply#8:** We will rewrite the sentence L29-30.

**Changes in the revised MS#8:** "In contrast, DIC was in majority in the form of dissolved $CO_2$ in groundwater. Concentrations of DIC were apparently diluted during high flow periods and were maximum during late summer period, when the overlying ecosystem was overall in heterotrophic conditions (i.e., $R_{eco}$>GPP)".

**Comment#9:** *"Can you please add a sentence on the major implication of the study?"*

**Changes in the revised MS#9:** We will add in the Abstract section a sentence on the major implication of our study: "On an annual basis leaching of terrestrial carbon to streams occurs as DIC and DOC in similar proportion; however, DOC export occurs in majority during short period of highest water table, whereas DIC export was relatively constant throughout the year. C export to the watershed represents a small portion of the net land carbon sink (about 1.5%)."

**Comment#10:** *"Introduction Line 59-60: Please add references"*

**Reply#10:** We will change this sentence and added references that focused on the factors controlling C export from land to streams in temperate forested catchment.

**Changes in the revised MS#10:** "The carbon dynamics in forest stream ecosystems results from the interaction between biogeochemistry, *i.e.* biological activity and retention mechanisms in soils, and hydrology, *i.e.* water infiltration in soils and drainage (Shibata et al., 2001; Kawasaki et al., 2005)."

**Comment#11:** *"Line 62: How so, please explain? No references, is this the first study to address this question?"*

**Reply#11:** Most studies on DIC/DOC exports from land to streams are based on soils and/or streams observations and not on groundwater sampling, that is restricted submarine (e.g., Santos et al., 2012), estuarine (e.g., Sadat-Noori et al., 2016), coastal floodplain (e.g., Atkins et al., 2013) and boreal lakes (e.g., Einarsdottir et al., 2017) environments, but rarely streams. The few studies on the role of groundwater as a carbon source to streams using direct measurements in groundwater are: 1/ Venkiteswaran et al (2014) who showed that DIC concentration in groundwater was higher than in streams because of fast CO2 degassing close to in stream waters close to resurgence points ; 2/ Kawasaki et al (2005) who showed that DOC export from groundwater is a minor component of the land sink because of DOC-removal mechanisms during vertical infiltration.

**Changes in the revised MS#11:** The Introduction section will be rewritten, describing the available information on carbon leaching from soils, but stressing the lack of studies that report direct carbon concentration measurements in groundwater.

**Comment#12:** *"I can understand the overall aim of the study, but it would be helpful if you could clearly layout research questions or even hypotheses."*

**Reply#12:** As mentioned previously (Changes in the revised MS#2), the objectives of our study will be clearly stated at the end of the Introduction section in the revised MS.

**Comment#13:** *"Methods Line 76: Several places you have used "per mill" symbol."*

**Reply#13:** We have removed the per mill symbol and express the basin slope in percent (%) in the revised MS.

**Comment#14:** *"Line 92: You are focusing on shallow groundwater, may be worth mentioning once for clarity?"*

**Reply#14:** In the revised MS, we will mention clearly both in the Materials and Methods and Discussion sections that we focused on shallow groundwater.

**Comment#15:** *"Line 97: Some abbreviations are not in the table 1; SOC?"*

**Reply#15:** In the revised MS, we will define SOC as soil organic carbon in Table 1.

**Comment#16:** *"Line 110: Sampled for what?*

**Reply#16:** In the revised MS, we will mention that we sampled shallow groundwater and first order streams for $pCO_2$, TA and DOC.

**Comment#17:** *"Line 166: How did you interpolate groundwater stages between two dates?"*

**Reply#17:** In the revised MS, we will explain clearly that values of water table depth were linearly interpolated between two dates.

**Comment#18:** *"Line 196: How did you estimate D, please explain. I am extremely confused about how you took the catchment scale response from a stream gauge, and after multiplying it by a factor of 2 assumed it is as discharge for the Bilos site?"*

**Reply#18:** We would like to apology for the poor and confusing description of how stream discharge was obtained in the present study. In fact, the detailed description of the hydrological approach

appears in a companion paper (Deirmendjian & Abril, submitted to Journal of Hydrology). When resubmitting the present paper, we will upload to the BG site the last version of this companion paper, in order to facilitate the reviewer's work. Because the two papers were written and submitted almost simultaneously, we did not realize that the description of hydrology was so poor in the present paper submitted to BGD. The companion paper describes the dynamics of DIC and its isotopic composition from groundwater to the outlet of the Leyre watershed (stream order 4) and contains a detailed analysis of river discharge data in streams orders 1 to 4. Our study took benefit from four calibrated gauging stations of the French water quality agency (with a daily temporal resolution), located on two second order streams, one third order stream and one fourth order stream (see revised Fig.1) (Deirmendjian, 2016). We also performed additional discharge measurements in first order streams. For each stream order, we calculated with a daily temporal resolution for a two years period the drainage factors (i.e., discharge divided by the corresponding catchment area, in $m^3$ $km^{-2}$ $d^{-1}$). We then determined the drainage enrichment factors α defined as the ratio between drainage factors of streams of successive orders. Because of the specific characteristics of the Leyre watershed with no surface runoff, we showed a regular increase in drainage factors (hence the drainage enrichment factors is > 1) between two streams of increasing orders, with enrichment factors relatively constant temporally. This allowed us a precise quantification of additional diffusive groundwater inputs in stream reach compare to that coming from the stream of inferior order. Our analysis leaded to the conclusion that monthly drainage values in first order stream was on average 2.3 times lower than that measured in fourth order stream. We will provide a copy of the submitted companion paper in order to help the review of the present paper.

**Changes in the revised MS#18:** In the revised MS, we will cite this companion paper and resume the hydrological method to make it fully understandable.

**Comment#19:** *"Line 214: With monthly sampling frequency, you will miss a lot of variation in DOC/DIC storage, especially during storms, no?"*

**Reply#19:** In fact, we believe that short storms would not have such crucial impact on the way we estimate carbon storage in groundwater based on monthly sampling frequency, because of the high permeability of the soil and the large storage capacity of water in the soil of the Leyre watershed.

In other types of environments (steeper and less permeable), DOC/DIC storage or export could be quickly affected by storms or pulsed hydrological events (e.g., Raymond and Saiers, 2010; Wilson et al., 2013). However, in the Leyre watershed, due to the combined effect of very low slope and high soil permeability, sudden hydrological events are buffered by temporary groundwater storage. Indeed, groundwater storage was strongly correlated with precipitation, whereas drainage was weakly correlated with precipitation (submitted Tab. 2). This buffer capacity of groundwater storage is illustrated in the revised figure 2: (i) river discharge increases in Feb. 2015, more than 2 months after the water table started to rise in Dec. 2014; (ii) river discharge remained constant between Sep. and Dec. 2015, although the water table was rising, but remaining at low stage; (iii) there was a lag time (about 3 months) between high precipitation and high river flow; (iv) with our monthly resolution we could appreciate the seasonal effect of DIC/DOC variations in shallow groundwater and streams

(revised Fig. 5), suggesting that the main factors that impact DIC/DOC variations were taking into account.

**Comment#20:** *" Line 233, Equation 7 will be followed by the equation 6?"*

**Reply#20:** In the revised paper, equation 6 will be 7 and equation 7 will be 6.

**Comment#21:** *"Line 246 In equation 8, how did you estimate Qmean, unclear?"*

**Reply#21:** $Q_{mean}$ is the mean river flow of first order streams in $m^3 d^{-1}$ between two sampling dates determined from our hydrological model.

It was estimated from the mean discharge of the Leyre River between two sampling dates divided by our correction factor of about 2.3 as explained above in Reply#18.

**Changes in the revised MS#21:** In the revised MS, we will describe how we estimate $Q_{mean}$ and we will mention the simple hydrological model citing our companion paper.

**Comment#22:** *"Results and Discussion- Too many ideas in a single paragraph and the main message is lost in the details. It is left to the reader to figure out the take home message. I would recommend re-writing results and discussion sections for better presentation of the key ideas"*

**Reply#22:** In the revised MS we will clarify both Results and Discussion sections, taking into account all comments made by both Referees.

See also reply#3 and #4

**Comment#23:** *"354: Given the study design, this sentence is a major overstatement"*

**Reply#23:** We agree with Referee 1 and rewrote the sentence.

**Changes in the revised MS#23:** "Our dataset –obtained during an 18 months long monitoring– enabled to understand how the water budget is impacted by the different hydrological parameters in the "Landes de Gascogne" area"

**Comment#24:** *"Line 460: Soil DOC in shallow layers cannot be mobilized by Precipitation? Lack of correlation between Soil DOC and P is not surprising as DOC exports from soil horizon is highly non-linear process"*

**Reply#24:** We are aware of this mistake and we are agree with Referee 1, that DOC exports from soil horizon are highly non-linear process and cannot be explain by Pearson's correlation that is a linear

correlation. The non-linear character of DOC and DIC mobilization from soil to groundwater and export to stream will be emphasized based on the new figure (see revised Fig. 4 below) which gives the DIC and DOC concentration in groundwater as a function of water table depth.

**Changes in the revised MS#24:** In the revised MS, we will rewrite the sentence as follows: "In addition, we never observed an increase of groundwater DOC concentration during base flow periods (when groundwater table is low) (see revised Fig. 5c). Thus, it seems that soil DOC in upper horizons cannot be mobilized by rainwater percolation and that saturation of the soil horizon is necessary to generate high DOC concentration in groundwater. This conclusion in in agreement with other works on groundwater DOC, e.g., Kawasaki et al. (2005) who show that retention mechanisms in soils during vertical infiltration strongly impact DOC exports to streams.

**Comment#25:** *"Line 490: Sentence unclear"*

**Reply#25:** We rewrote the sentence.

**Changes in the revised MS#25**: "Water table depth impacts the watershed pressure hydraulic head driving the drainage, as well as the concentrations of DIC and DOC in the groundwater."

**Comment#26:** *"Tables and Figures. Table 5. It is good to provide conceptual table and later figure 7, but please remember you can only interpret so much from your limited study design."*

**Reply#26:** We took this comment into account.

**Changes in the revised MS#26:** The figure 7 and the conceptual table of the submitted MS will be replaced by another figure (see new Figure 7 below). The new figure 7 of the revised MS allows describe in a qualitative way processes and fluxes at the groundwater/stream interface without excessive interpretation.

**Comment#27:** *"Fig1. If possible, please show locations of the rain gauge and flux tower (or location of Eddy covariance measurements) on the map"*

**Reply#27:** In the revised MS, we will indicate on the legend that rain gauge and flux tower are located at the Bilos site.

**Changes in the revised MS#27:** "Legend revised Fig. 1: Map of the Leyre watershed with topography showing the location of the gauging stations (GL, PL, GAR and BR for the Grande Leyre, the Petite Leyre, the Grand Arriou and the Bourron, respectively), the Bilos site as well as the locations of the other sampled groundwater and first order streams. Rain gauge and Eddy tower are located at the Bilos plot."

**Comment#28:** *"Fig4: There is no gray side bar in the figure. Please revise caption or add the bar to the figure. How did you estimate error bars, _1 standard deviation?"*

**Reply#28:** The error bars were standard deviation of the 6 first order streams. In the revised MS, we will modify the legend. The side bars representing the different hydrological periods were added (see below).

**Changes in the revised MS#28:** Please see revised Fig. 5 below.

**Comment#29:** *"Fig7. The texts within the boxes are bit small, if possible please increase their font sizes."*

**Reply#29:** In the revised MS, this figure will be deleted and replaced by a new Fig. 7 (see below)

References cited

[revised manuscript text omitted]

**New Figure 8: concentration of carbon in shallow groundwaters and fluxes of carbon at the vegetation-groundwater-stream-atmosphere over the 4 hydrological periods. Fluxes of carbon are in mmol m$^{-2}$ d$^{-1}$. Concentrations of carbon ([DOC] and [DIC]) are in mol m$^{-3}$. Error bars represent the standard deviation of the different hydrological periods.**

---

## Author Comment (AC2) · 12 Jul 2017

Dear Reviewer,

First of all we would like to thank you for the time and effort you spent on reviewing our MS. We very much appreciate your comments that clearly identify some parts of the submitted MS that need more effort, clarification and re-organization. We will try to address all the questions and comments you raised and we are convinced that the revised MS will be improved significantly.

Best regards,

Loris Deirmendjian & Co-authors

**Comment#1:** *"Interesting, significant work, but presentation is poor and unacceptable. One of the reason for this is insufficient reviews of similar studies. Although the studies in which DOC/DIC exports are compared with NEE measured by eddy covariance method in the same site may be limited, there are many studies discussed about DOC/DIC exports and their mechanisms."*

**Reply#1:** We thank the Referee 2 for his/her constructive comment concerning the organization and presentation of our MS. This general evaluation is quiet consistent with that expressed by Referee 1 and for that reason we will be able to provide a revised MS that will satisfy all these comments. We clearly realize that the submitted paper suffered from poor organization and presentation and needed a complete re-organization and a more thorough review of similar studies in forested catchments. However, we would like to stress the general scarcity of studies that report DIC and DOC concentration in groundwater, and even less studies that compare groundwater and streams. We have performed a thorough review of similar study in forested and other type of catchments, which reveals that, the few available works that estimates lateral export of carbon are based: (1) on observations in soil water in the unsaturated zone combined with soil water model (Öquist et al., 2009; Kindler et al., 2011; Leith et al., 2015), (2) on differences in the flux between upper and lower stream reaches (Shibata et al., 2001, 2005), or (3) on observations in stream water combined with stream discharge (Dawson et al., 2002; Billett et al., 2004; Dinsmore et al., 2010; Olefeldt et al., 2013). We found few works that reported DIC/DOC concentrations and dynamics in groundwaters (saturated zone). Kawasaki et al. (2005) reported DIC/DOC dynamics in groundwaters but this study is out of the scope of our watershed with sandy podzols as it focused on a watershed with granite rocks overlied by cambisols. Venkiteswaran et al (2014) reported DIC concentrations in groundwaters but this paper focus on degassing in streams based on an isotopic model. Artinger et al (2000); Baker et al (2000) and Shen et al. (2015) reported DOC dynamics and concentrations in groundwaters but these works focus on the characterization and/or metabolism of groundwater DOC and not on carbon export to streams.

**Changes in the revised MS#1:** We will cite these references in the revised MS, stressing their different scientific objectives and approaches. We will put important effort in improving our revised MS to address the questions of presentation, description of the state of the art, and discussion in the light of previous works. We will modify the Introduction section and profoundly rework the text of the Results and Discussion sections of our revised MS that will include 2 new figures (Fig. 7 and 8 in the revised MS) and 5 revised figures (Fig. 1, 2, 4, 5, 6 in the revised MS) (see new figures at the end of this document). The figure 3 of the submitted MS did not change. The figure 6 and 7 of the submitted MS were removed (see below).

**Comment#2:** *"The result section is hard to catch up. It should be rearranged to show more clearly to guide the readers for your discussion."*

**Reply#2:** We agree with the need of better organizing the Results section, a comment also shared by the Referee 1.

**Changes in the revised MS#2:** In the revised MS, we will split the Results section in five paragraphs: (i) water mass balance; (ii) carbon net ecosystem exchange; (iii) dissolved carbon in groundwater; (iv) dissolved carbon in first order streams; (v) hydrological carbon fluxes.

Firstly, in the 3.1 section, based on hydrological and eco-physiological criteria we will distinguish 4 different hydrological periods within the sampling period (Jan. 2014-Jul. 2015): HF, GS, LS and EW periods, respectively for high flow, growing season, late summer and early winter periods (see revised Fig. 2). Then, results will be described according to these 4 hydrological periods in the revised text as well as in the different revised figures, which we believe, will lead to greater clarity and understanding.

Secondly, we will modify Fig.4 (of the submitted MS) by separating the distribution of DIC and DOC concentrations in groundwater as a function of water table depth (panel A), from the time-course of concentrations (new Fig. 4 and 5 in the revised MS) (see below). We will also add new Figures 7 and 8, which describe a conceptual model of the groundwater-stream interface and the different measured and calculated carbon fluxes between the atmosphere, the groundwater and the streams during these 4 different periods and, respectively (see new Fig. 7 and 8 at the end of this document). These two new figures allowed us to better discuss the seasonality of carbon fluxes in interaction with hydrological and biogeochemical processes without excessive interpretations.

Thirdly, as suggested by both referees, results of the Pearson's correlations will be much less emphasized in the revised version of the MS, as they appeared too simplistic to fully explain biogeochemical and hydrological processes controlling dissolved C concentrations and fluxes.

**Comment#3:** *"The discussion section is also need to rearrange. I suppose that the contents in this is version of manuscript may be true, but may be incorrect at this moment."*

**Reply#3:** In the revised MS, we have split the Discussion section in three paragraphs: (i) water mass balance; (ii) soil carbon leaching to groundwater; (iii) carbon transfer at the groundwater-stream interface.

In addition, we will clarify the Discussion section and add more literature and discussion about DIC/DOC export from land to surface waters. However, we also stress the lack of carbon studies in groundwater, the knowledge gap at the soil-groundwater interface, and how our study helps in feeling these gaps.

We will also modify significantly our figures by splitting Fig. 4 (panel a and panels b&c of the submitted MS) into two separate figures (revised Fig.4 and 5). Indeed, we believe that the opposite evolution of groundwater DIC and DOC with water table is a first key result of our work that needs to be more highlighted. We will also show additional discharge data in first, second, and third order streams that will allow a much better description of the simple hydrological model used in the watershed.

**Comment#4:** *"L.59 '... generally'. Here, the authors must refer some studies. Moreover, clarify how different this study from these studies?"*

**Reply#4:** We will change this sentence and added references that focused on the factors controlling C export from land to streams in temperate forested catchment.

**Changes in the revised MS#4:** "The carbon dynamics in forest stream ecosystems results from the interaction between biogeochemistry, *i.e.* biological activity and retention mechanisms in soils, and hydrology, *i.e.* water infiltration in soils and drainage (Shibata et al., 2001; Kawasaki et al., 2005)."

**Comment#5:** *"L.74. The Leyre watershed is very large, but the piezometers are only three. Consider and clarify whether or how the data shown here represent whole the watershed. In other words, do the results in Bilos site (0.6km2) represent the whole Leyre watershed?"*

**Reply#5:** In the Leyre watershed, there are 3 types of soil: wet Landes, dry Landes and mesophyl Landes (Augusto et al., 2006; Jolivet et al., 2007). Each type was characterized with different amplitude of water table depth. In the upper parts of the watershed, as well as near watercourses, the water table is always more than 2 m deep: it is the dry Landes. In the lower parts, or in the vast interfluves, the groundwater is close to the surface of the soil in winter (0.0-0.5 m depth) and generally remains close to it, even in summer ( 1.0-1.5 m): it is the wet Landes. The mesophyl Landes corresponds to the intermediate situation. In the Leyre watershed, the dry Landes, the mesophyl Landes and the wet Landes represents respectively 17%, 36% and 47% of the watershed area (Augusto et al., 2006). This spatial soil distribution is representative of the "Landes de Gascogne" forest massif (Trichet et al., 1999). According to the depth and amplitude of the water table, our three piezometers were representative of dry Landes (Piezometer 2), mesophyll Landes (Piezometer 3) and a situation between mesophyll and wet Landes (Piezometer Bilos).

**Changes in the revised MS#5:** We agree with the reviewers that the question of representatively of piezometers must be addressed when extrapolating fluxes at the scale of the watershed. In the revised MS, we will estimate C export from shallow groundwaters to first order streams with the data of the 3 piezometers (not only Bilos), by calculating an export with a mean weighted (by surface) concentration of carbon in the 3 shallow groundwaters. This procedure will improve the mass balance at the scale of the watershed; however, we will still discuss the limits of the approach in the revised MS, providing maximum and minimum C export for the three piezometers.

**Comment#6:** *"L. 80 & 90 XIXth Change to Arabic"*

**Changes in the revised MS#6:** We changed to Arabic.

**Comment#7:** *"L. 91.Consequently...' References about hydrology in the Leyre watershed must be sited around this sentence. If the authors does not cite any studies here, the descriptions are suspicious."*

**Reply#7:** We would like to apology for the poor and confusing description of how stream discharge was obtained in the present study. In fact, the detailed description of the hydrological approach appears in a companion paper (Deirmendjian & Abril, submitted to Journal of Hydrology). When resubmitting the present paper, we will upload to the BG site the last version of this companion paper, in order to facilitate the reviewer's work. Because the two papers were written and submitted almost simultaneously, we did not realize that the description of hydrology was so poor in the present paper submitted to BGD. The companion paper describes the dynamics of DIC and its isotopic composition from groundwater to the outlet of the Leyre watershed (stream order 4) and contains a detailed analysis of river discharge data in streams orders 1 to 4. Our study took benefit from four calibrated gauging stations of the French water quality agency (with a daily temporal resolution), located on two second order streams, one third order stream and one fourth order stream (see revised Fig.1) (Deirmendjian, 2016). We also performed additional discharge measurements in first order streams. For each stream order, we calculated with a daily temporal resolution for a two years period the drainage factors (i.e., discharge divided by the corresponding catchment area, in $m^3 \ km^{-2} \ d^{-1}$). We then determined the drainage enrichment factors α defined as the ratio between drainage factors of streams of successive orders. Because of the specific characteristics of the Leyre watershed with no surface runoff, we showed a regular increase in drainage factors (hence the drainage enrichment factors is > 1) between two streams of increasing orders, with enrichment factors relatively constant temporally. This allowed us a precise quantification of additional diffusive groundwater inputs in stream reach compare to that coming from the stream of inferior order. Our analysis leaded to the conclusion that monthly drainage values in first order stream was on average 2.3 times lower than that measured in fourth order stream. We will provide a copy of the submitted companion paper in order to help the review of the present paper.

In addition, we should mention that the Leyre watershed is a very flat coastal plain with almost no slope (mean slope is less than 0.125%) (Jolivet et al., 2007), soils are sandy with allover no clay minerals (i.e., very low soil water retention) (Augusto et al., 2006) and permeable (i.e., hydraulic

conductivity is about $10^{-4}$ m s$^{-1}$, Corbier et al., 2010). Combining the above factors the infiltration of rain water was fast (around 50 to 60 cm h$^{-1}$ on average) (Vernier and Castro, 2010) and surface runoff cannot occur. To support this, (i) our water budget calculated from drainage (that considers only shallow groundwater drainage) was fairly consistent (Fig. 3, end of this document); (ii) there was a lag time (about 3 months) between high precipitation and high river flow (revised Fig. 2, end of this document).

**Changes in the revised MS#7:** In the revised MS, we will cite our companion paper and resume the hydrological method to make it fully understandable. We will also provide additional information on local hydrology including references on the subject.

**Comment#8:** *" L.103. Although soil preparation, fertilization and seeding was done in 2005, the referred paper was published in 2003. Why ?"*

**Reply#8:** We would like to apology for this imprecise and confusing statement. In fact, in December 1999, cutting of the 50 years old forests began with on one quarter of the plot (15 ha) but it was delayed due to a windstorm (Kowalski et al., 2003). Following clear-cutting in 1999, the site was ploughed to 30 cm depth and fertilized with 60 kg P per ha in 2001 (Moreaux et al., 2011). Measurements of eddy covariance fluxes at this site started in 2000 (Moreaux et al., 2011). In November 2004, the site was divided into two parts, which were seeded with maritime pine (Pinus pinaster) with a 1-year lag, in 2004 and 2005, respectively, tree rows being spaced at 4 m (Moreaux et al., 2011).

**Changes in the revised MS#8:** The above information will be added in the revised MS.

**Comment#9:** *" L.180. What is the 'CO2 SYS'? No information"*

**Reply#9:** CO2 sys (Lewis et al., 1998) is a EXCEL Macro spreadsheet that calculates the concentrations of all species of the carbonate system from two measured parameters (in our study we calculated DIC from $pCO_2$ and alkalinity) temperature, and pressure.

**Changes in the revised MS#9:** Reference Lewis et al (1998) will be cited in the revised MS.

**Comment#10:** *" Result. Too complicated. I highly recommend the authors to reconsider what are your main points."*

**Reply#10:** As we mentioned above (in Changes in the revised MS#2), we will rearrange and clarified the Result section.

**Comment#11:** *"L.376. 'groundwater uptake'. This term is suspicious. Generally, many plants including pine trees use the soil water within unsaturated zone, and cannot survive if the roots are immersed in groundwater, saturated zone"*

**Reply#11:** Referee 2 is right, and this is the reason why the network of drainage ditches was created by foresters in order to accelerate the evacuation of the water in excess when the groundwater level rises. In addition, Bakker et al (2006) and (2009) showed that pine roots in the Landes of Gascogne area are confined to the first meter of the soil, probably because of winter anoxia related to the groundwater saturation. However, pine trees exhibit water uptake from the water table and the unsaturated zone (Vincke and Thiry, 2008)**.**

**Changes in the revised MS#11:** In the revised MS, we will rewrite the sentence as follows: "The water table depth (and thus the groundwater storage) is mostly impacted by evapotranspiration of pine trees, as attested by (i) the correlation between groundwater storage and evapotranspiration (submitted Tab.2) and (ii) this is agree with Vincke and Thiry (2008) who found that water table uptake contributes to 61% of the evapotranspiration for an experimental Scots Pine plot in a flat area of Belgium and (iii) with Guillot et al (2010) who found that groundwater contribution to the evapotranspiration was 50% in the Bilos pine plot.

**Comment#12:** *"L. 378- and Table 2. The discussions are generally based on the Pearson's correlation coefficient. However, the real hydrological and biogeochemical phenomena cannot be sufficiently described by this kind of simple value; for example, how do you explain the time lag among precipitation, response of groundwater table, and drainage? "The relationship between the groundwater storage and drainage in your site (L.381) is unclear in your consideration, but I suppose that it means the nonlinearity between these parameters, and it cannot be shown by the Pearson's correlation coefficient."*

**Reply#12:** The lag time between precipitation and groundwater storage is short at our study site, as attested by the strong linear relationship between GWS and P (submitted Tab. 2). This is consistent with the sandy texture of soils with a high proportion of coarse sands (between 60 and 70%) in the Landes de Gascogne (Augusto et al., 2006) which makes the infiltration of rain water fast (around 50 to 60 cm h$^{-1}$ on average) (Vernier and Castro, 2010). The correlation between GWS et P is also consistent with findings of Alley et al (2002) who highlight that groundwater recharge can occur in response to individual precipitation in regions having shallow groundwater table.

However, there is a lag time between groundwater storage and drainage, attested by the nonlinear relationship between GWS and D (submitted Tab. 2). This lag time is due to the combined effect of very low slope and high soil permeability that prevent pulsed hydrological events. In the Leyre watershed, shallow groundwaters acts as a buffer, the river flow being mostly controlled by water table depth and its capacity to store or export water. This buffer capacity of groundwater storage is illustrated in the revised figure 2: (i) river discharge increases in Feb. 2015, more than 2 months after the water table started to rise in Dec. 2014 (revised Fig. 2); (ii) river discharge remained constant between Sep. and Dec. 2015, although the water table was rising, but remaining at low stage (revised

Fig. 2); (iii) there was a lag time (about 3 months) between high precipitation and high river flow (revised Fig. 2)

**Changes in the revised MS#12:** The above information will be added in the revised MS. In addition, the revised figure 4 that shows DIC and DOC concentrations in groundwater as a function of water table depth will allow us to stress more the non-linearity of the processes mobilizing dissolved C from soil to groundwater and then from groundwater to streams.

**Comment#13:** *"The authors also mentioned about this at L.382, but why this occurs in the flat topography of the watershed? Any references? I think this occurs not only in flat watershed, but also in steep watershed. Again, many discussions are not supported by reliable previous studies, previous knowledge.*

**Reply#13:** Transfer of precipitation to rivers involves temporary water storage in reservoirs where different residence times influence the hydrological cycle (*e.g.* Oki and Kanae, 2006). The time of travel in the soil system depending on the spatial temporal gradient of hydraulic head, hydraulic conductivity, and porosity of the system (Alley et al., 2002; Ahuja et al., 2010). In lowland watershed, with overall a small hydraulic gradient, we expected a residence time in groundwater longer than in a steep watershed.

**Comment#14:** *"L.388-395. The discussion about (annual?) soil water storage is not clear for me. The authors mentioned that it was larger in 2015 (126mm) than in 2014 (71mm). Is this correct? The annual rainfall was much smaller in 2015, and the soil water storage will be smaller as well as the groundwater storage. If the authors can show the data of soil water and/or hydraulic head, we can get more reliable information."*

**Reply#14:** We totally agree with this comment. We do not have soil water data or hydraulic gradient data at the Bilos site. Consequently, as the Referee 2 mentioned, some discussion on water storage cannot be done.

**Changes in the revised MS#14:** We will remove this paragraph from the revised MS. Water budget is secondary for our revised MS which is focused on carbon.

**Comment#15:** *" L. 402. Insert period after 'bicarbonates'."*

**Reply#15:** We inserted period.

**Comment#16:** *" L. 463. Discussion of this paragraph is strange""L. 470 'Thus, when the forest ecosystem is a source of CO2 for the atmosphere, it is also a source of CO2 for the underlying groundwater.' ... Even under the drought condition or even when the ecosystem act as a sink of CO2, below ground part of the vegetation 'only' act as the CO2 source; respiration by root always occur.*

*Moreover, degradation of organic carbon, including DOC always occur. It's a respiration by microorganisms"*

**Reply#16:** Indeed, we apologize for the confusing formulation in the submitted MS. The Referee 2 is right when she/he mentions that respiration always occurs in soils.

**Changes in the revised MS#16:** We will change the text of our revised MS as follows: "thus, the late summer period, when the forest ecosystem is a source of $CO_2$ for the atmosphere, also corresponds to a maximum in $CO_2$ concentration in groundwater and thus a maximum contribution of soil respiration to groundwater DIC" To our best knowledge this is another original finding of our work.

**Comment#17:** *"L.482 and other parts. Mean residence time, define by S/flux in this manuscript, should be reword as 'apparent turnover time'. In addition, it was calculated between two sampling days. This method cannot consider the time lag between the storage and output flux, as same as mentioned above. The Pearson's correlation coefficient cannot explain this"*

**Reply#17:** We will take this comment into account in the revised MS, by changing term "mean residence time" to "apparent turnover time" and defining precisely how it was calculated. It is worth to note that in our MS, we use the apparent turnover times only to support the fact that during the growing season periods, DOC apparent turnover time in groundwater is long enough to assume that respiration of this DOC accounts for a large part of the increase in groundwater DIC.

**Changes in the revised MS#17:** This information does not require a Figure, and figure 6 of the submitted MS was removed.

**Comment#18:** *"L. 503. The comparison with peat systems in less meaningful. Should be compared to similar ecosystem with your site. Moreover, I agree with the contents of the four referred papers (L.503-507), but how related them with your results? As I mentioned above, your discussion does not consider about the time lag, or times for decomposition and transport. The referred studies essentially mentioned about this"*

**Reply#18:** Indeed, the reference concern peatland ecosystems which might behave very differently from our well-drained sandy watershed.

**Changes in the revised MS#18:** However, in the revised MS we will add other references (Shibata et al., 2005; Jonsson et al., 2007; Dinsmore et al., 2010; Kindler et al., 2011). These studies all show the importance of lateral export of DOC and DIC from different terrestrial systems and that could potentially represent 40-100% of the NEE in peatland systems (Billett et al., 2004; Dinsmore et al., 2010), 10% in boreal systems (Jonsson et al., 2007) , 28% in forests, grasslands, and croplands across Europe (Kindler et al., 2011) and 2% in Japan temperate deciduous forest (Shibata et al., 2005). We will discuss the fact that the proportion of NEE exported laterally by varies a lot depending on ecosystem types. We would like to discuss the fact that in other types of ecosystems, a part of the NEE may be lost by other processes such as litter fall, fine roots turnover and root exudates, even if we cannot relate these processes with our results. We will also highlight the importance to extend this

investigation to other landscapes, climatic zones, soil types, vegetation and hydrological regimes, that could considerably improve estimates of carbon budgets of terrestrial ecosystems.

**Comment#19:** *" L. 508. What's the meaning of this sentence here?"*

**Reply#19:** We will change this sentence to "As in groundwaters, DOC and DIC in first order streams were anti-correlated, suggesting that C dynamic in first order streams was mostly impacted by groundwater inputs".

**Comment#20:** *" 'reported' ... Need appropriate citation. 'elsewhere'... ??? Where? Too irresponsible!!"*

**Reply#20:** In the revised MS, we will cite the references Dawson et al. (2002), Striegl et al. (2005), Raymond and Saiers (2010) and Alvarez-Cobelas et al. (2012) in order to compare our study site with other ecosystems and reinforce our statements.

**Comment#21:** *"L. 512. As I mentioned at L. 74, I wonder how the authors can show that the observed groundwater is representative of whole the watershed. I think the authors cannot discuss about the spatial variation of data. This is one of the weak point of this paper"*

**Reply#21:** In the Leyre watershed, there are 3 types of soil: wet Landes, dry Landes and mesophyl Landes (Augusto et al., 2006; Jolivet et al., 2007). Each type was characterized with different amplitude of water table depth. In the upper parts of the watershed, as well as near watercourses, the water table is always more than 2 m deep: it is the dry Landes. In the lower parts, or in the vast interfluves, the groundwater is close to the surface of the soil in winter (0.0-0.5 m depth) and generally remains close to it, even in summer (1.0-1.5 m): it is the wet Landes. The mesophyl Landes corresponds to the intermediate situation. In the Leyre watershed, the dry Landes, the mesophyl Landes and the wet Landes represents respectively 17%, 36% and 47% of the watershed area (Augusto et al., 2006). This spatial soil distribution is representative of the "Landes de Gascogne" forest massif (Trichet et al., 1999). According to the depth and amplitude of the water table, our three piezometers were representative of dry Landes (Piezometer 2), mesophyll Landes (Piezometer 3) and a situation between mesophyll and wet Landes (Piezometer Bilos). Combining the above factors, with a relatively homogenous lithology, topography and slope as well as simple hydrological functioning all over the Leyre watershed, the representativity of our piezometers is quite good because we selected contrasting study site representative of the diversity of the ecosystem.

See also Reply#5.

**Changes in the revised MS#21:** We agree with the reviewers that the question of representatively of piezometers must be addressed when extrapolating fluxes at the scale of the watershed. In the revised MS, we will estimate C export from shallow groundwaters to first order streams with the data of the 3 piezometers (not only Bilos), by calculating an export with a mean weighted (by surface)

concentration of carbon in the 3 shallow groundwaters. This procedure will improve the mass balance at the scale of the watershed; however, we will still discuss the limits of the approach in the revised MS, providing maximum and minimum C export for the three piezometers.

**Comment#22:** *"L.513-. 'Also, ...'. I agree with the first part of this sentence (but need some references about this phenomena), but cannot agree with the second part. Why the absence of correlation support the phenomena? The correlation is just a correlation; it cannot attest the phenomena. Comparison of concentration is different from the correlation"*

**Reply#22:** We will take this comment into account and rewrite the sentence as suggested:

**Changes in the revised MS#22:** This might be due partly because groundwater DOC is quickly respired before reaching the stream. Also, as groundwater DOC enters the superficial river network through drainage it might be rapidly recycled by photo-oxidation (Macdonald and Minor, 2013; Moody and Worrall, 2016) or respiration within the stream (Roberts et al., 2007; Hall Jr et al., 2016)/

**Comment#23:** *"L. 517. (Righi and Wilbert, 1984) This reference is too old. The studies about DOC quality is making steady progress. I think the authors can find more up-to-date studies to support your findings.?"*

**Reply#23:** We added a more recent reference (Augusto et al., 2006) in the revised MS.

**Comment#24:** *"L. 521. There are no data, analysis, and discussion about the relationship between the topography and DOC concentration. How and why the difference of DOC concentration between three piezometers was occurred by the effect of topography"*

**Reply#24:** We wanted to highlight the fact that the export of DOC is impacted by the water table depth. Indeed (i) soil DOC seems to be mobilized only when soil organic horizons are saturated with water; and (ii) water table depth drives the watershed pressure hydraulic head driving the drainage. The water table depth might be influenced with local hydrological heterogeneities or with local topography effect resulting from different soil types. In the Leyre watershed, there are 3 types of soil: wet Landes, dry Landes and mesophyl Landes (Augusto et al., 2006; Jolivet et al., 2007). Each type was characterized with different amplitude of water table depth. In the upper parts of the watershed, as well as near watercourses, the water table is always more than 2 m deep: it is the dry Landes. In the lower parts, or in the vast interfluves, the groundwater is close to the surface of the soil in winter (0.0-0.5 m depth) and generally remains close to it, even in summer ( 1.0-1.5 m): it is the wet Landes. The lmesophyll Landes corresponds to the intermediate situation. In the Leyre watershed, the dry Landes, the mesophyl Landes and the wet Landes represents respectively 17%, 36% and 47% of the watershed area (Augusto et al., 2006). This spatial soil distribution is representative of the "Landes de Gascogne" forest massif (Trichet et al., 1999). According to the depth and amplitude of the water table, our three piezometers were representative of dry Landes (Piezometer 2), mesophyll Landes (Piezometer 3) and a situation between mesophyll and wet Landes (Piezometer Bilos).

See also reply#5 and #21.

**Changes in the revised MS#24:** We will calculate the DOC (and DIC) export rates from each piezometers and account for the spatial heterogeneity induced by topography by calculating an export with a mean weighted (by surface) concentration of carbon in the 3 shallow groundwaters.

**Comment#25:** *" L. 525. The authors have mentioned about the possibility of photodegradation (or photo oxidation) of groundwater DOC at L. 514. However, the authors also mentioned as 'DOC was not labile, and not degraded in the superficial river network' during base flow period. These two comments make me confuse. Generally, the base flow condition occurs in fine (or no rain) days. Why the photodegradation does not occur under base flow condition? Which comment is true for you (and for us)?"*

**Reply#25:** During high flow periods, the higher DOC concentration in shallow groundwaters than in first order streams could originates from 2 phenomena: (i) oxidation of DOC in the groundwater itself, which leads to a simultaneous increase in groundwater DIC (see revised Fig. 5b-c and new Fig. 7) (ii) respiration and photo-degradation of groundwater-derived DOC in first order streams. However, according to the mean stock of DOC ($S_{DOC}$), the mean export of DOC ($DOC_{ex}$) and the respiration rates of 93 mmol m$^{-2}$ d$^{-1}$ (L.444 of the submitted MS) during HF periods, the second phenomenon seems to be less significant than the first because about 60% of the stock of groundwater was respired directly in the groundwaters and only about 4% was exported to streams during HF periods.

During the other hydrological periods, DOC concentrations in shallow groundwaters and in first order streams are very similar and stable, suggesting that DOC behaved conservatively. This was consistent with findings of Schiff et al. (1997) who found that a small temperate basin had wide range in 14C-DOC, from old groundwater values at base flow under dry basin conditions to relative modern values during high flow or wetter conditions.

**Changes in the revised MS#54:** We will carefully discuss these issues and cite theses references in the revised MS.

References cited

[revised manuscript text omitted]

**Revised Fig. 2: Seasonal variations of hydrological parameters in the Leyre watershed. (a) Discharge of the Leyre River (GL), Petite Leyre river (PL), Grand Arriou (GAR) river and Bourron (BR) river associated with water table at the Bilos site; (b) Eco-physiological parameters (NEE, GPP, $R_{eco}$) estimated at the Bilos site; (c) Monthly precipitation (P), drainage (D), evapotranspiration (ETR) and groundwater storage (GWS) at the Bilos site. HF, GS, LS and EW represent respectively high flow, growing season, late summer, and early winter periods.**

[Figure]

**Figure 3: Monthly water mass balance (see section 2.5) at the Bilos site for 2014-2015. Pearson coefficient R = 0.85, p-value < 0.001. Blue points represent months where GWS (Mar. 2014, Apr. 2014, Mar. 2015, Apr. 2015, Jun. 2015, Jul. 2015) is extremely negative (see Fig. 2). These blue points are further away from the 1:1 Line than the other months (represented in black). The drainage of the Leyre River is delayed compared to the drainage of Bilos plot. Thus, when the loss of groundwater is extremely high (GWS negative), calculated D do not correspond to measured GWS. Hence, we expected more mistakes when GWS is extremely negative.**

[Figure]

**Revised Fig. 4: Concentration of DIC and DOC in groundwaters as a function of water table**

[Figure]

**Revised Figure 5: (a) Discharge of the Leyre (GL), Petite Leyre (PL), Grand Arriou (GAR) and Bourron (BR) rivers associated with dwater table at the Bilos site Temporal variations throughout the sampling period of (b) DIC in groundwater (Bilos and the two other piezometers) and DIC in the 6 first order streams (medium dashed line; errors bars represent standard deviation of the 6 first order streams) and (c) DOC in groundwater (Bilos and the two other piezometers) and DOC in the 6 first order streams (medium dashed line; errors bars represent standard deviation of the 6 first order streams). HF, GS, LS and EW represent respectively high flow periods, growing season periods, late summer periods and early winter periods.**

[Figure]

**Revised Figure 6: Temporal variations throughout sampling period of (a) ecological parameters at the Bilos site (*GPP*, *R* and *NEE*), here *GPP* is represented negative (b) storage of DIC and DOC in groundwater at the Bilos site and (c) export of DIC and DOC throughout Bilos groundwater and degassing of CO₂ in first order streams. HF, GS, LS and EW represent respectively high flow periods, growing season periods, late summer periods and early winter periods.**

[Figure]

**New Fig. 7: Conceptual model at the vegetation-soil-groundwater-stream interface of the Leyre catchment. OH, WT, D are organic horizon of the soil, groundwater table and drainage, respectively. Hydrobiogeochemical processes are represented in medium dash black arrows. Carbon export are represented in full arrows, the thickness of the arrow provide qualitative information on the flux. . HF, GS, LS and EW represent respectively high flow periods, growing season periods, late summer periods and early winter periods.**

[Figure]

**New Figure 8: concentration of carbon in shallow groundwaters and fluxes of carbon at the vegetation-groundwater-stream-atmosphere over the 4 hydrological periods. Fluxes of carbon are in mmol m$^{-2}$ d$^{-1}$. Concentrations of carbon ([DOC] and [DIC]) are in mol m$^{-3}$. Error bars represent the standard deviation of the different hydrological periods.**

---

## Author Comment (AC4) · 12 Jul 2017

*Comments to the Author:*

*Dear Authors,*

*The contents of your field observation are very interesting, and I think that the manuscript reached to the level for discussion and review process, although the results and discussions were bit too descriptive. More discussions towards the implications and generalizations of the finding from this particular research. For this, several other references might be useful. But, I would like to proceed the review process. Please consider above points when you make revisions based on the reviewer's comments in the next step.*

*Regards,*

*Nobuhito Ohte, Associate Editor*

Dear Nobuhito Ohte,

First, we would like to thank for the time you spent on our submitted MS, and for your overall positive and constructive evaluation of our submitted MS. Second, as also mentioned by both referees, we clearly realized that some parts of our submitted MS were too descriptive and suffer from poor organization and need a more thorough review of similar studies in forested catchments. Consequently, we will put important effort in improving our revised MS to address the questions of presentation, description of the state of the art, and discussion in the light of previous works. We will modify the Introduction section and profoundly rework the text of the Results and Discussion sections of our revised MS that will include 2 new figures (Fig. 7 and 8 in the revised MS) and 5 revised figures (Fig. 1, 2, 4, 5, 6 in the revised MS) (see replies for referees). We provide our replies to both referees and we will try to address all the questions and comments they raised and we are convinced that the manuscript will be improved significantly. In the following documents, we sorted all comments/questions (Comment#x) and provide for each an answer (Reply#x) and sometimes the future changes in the revised MS (Changes in the revised MS#x). However, the revised MS is not yet ready.

Best regards,

Loris Deirmendjian & Co-authors

---

## Author Response (AR1)

Dear Editor,

On behalf of my co-authors, I am pleased to submit our revised MS entitled: "Hydrological and metabolic controls on
5   dissolved carbon dynamics in groundwater and export to surface waters in a temperate pine forest". The MS has been
intensively rewritten following the recommendations of the two referees. The referees pointed out the poor clarity and
organization in our first submitted version and a superficial reference to the literature. For these reasons, the introduction,
results, discussion and conclusion sections were almost entirely re-written. We also added in the materials and methods
section a more complete description of how stream discharge was obtained, and how drainage was modeled in the study
10   catchment. We also performed a more thorough review (81 references were added and 18 were removed as they were
useless) of similar studies in forested catchments that helped to discuss in more details the biogeochemical processes
occurring at the vegetation-groundwater-stream-atmosphere continuum.

In this revised version of the MS, we also modified the figures and tables in order to improve the presentation, help the
15   discussion and provide some additional quantitative information on water and carbon budget necessary for the discussion. To
make our paper absolutely clear, two additional tables that describe in detail the sampling periods in each sampling plot were
added as supplementary material. In order to help the description of material and methods, and the discussion, we added to
the figure 1 the topography (instead of land cover) and the locations of the different gauging stations. In the figure 2 we
added the eddy covariance parameters at the Bilos plot as well as the different hydrological periods. The figure 3 did not
20   change. We split former figure 4, in two separate figures (4 and 5). The new figure 4 presented the opposite evolution of DIC
and DOC with water table and amplitude one key result of our work. The new figure 5 presented the temporal dynamics of
DIC and DOC in groundwater and surface water at all stations. We deleted the figure 5, 6, 7 of the first submitted version
that suffered from redundancy with other figures, from poor clarity or from excessive interpretations. Instead, we added new
figures 6 and 7. Figure 6 presents the mean stocks of DIC and DOC in groundwater during the different hydrological
25   periods, and figure 7 is a conceptual model that synthesizes the biogeochemical processes occurring at the vegetation-soil-
groundwater-stream interface of the Leyre catchment, as highlighted by our results.

We are looking forward the new evaluation of our revised paper.

30   Best Regards,

Loris Deirmendjian and Co-authors

---

## Author Response (AR2)

Dear Editor,

Please find below our responses to the two referees together with a description of the changes made in the MS. We took care to satisfy most of the comments by the two referees and to improve the quality of the MS. In this revision, we displace some statements from the Results section to the Material and Methods section, we rewrote entire sections of text of discussion (mainly in the section 4.2) and we rewrote as well the entire conclusion in a more generalized way as suggested previously by the two referees. From the last submitted version of our MS we modified Figure 1, 2, 4, 5, 6 (basically we edited colour in the figures) and 7 (we added more details in our conceptual model). From the last submitted version of our MS we deleted Table 1 (by the way, we deleted all the abbreviations in the MS except DIC and DOC) and we modified Table 2 (we edited all units in mm $d^{-1}$), 3 (we added header) and 5 (we added the number of samples for each hydrological period). We also shortened the MS (in font 12, 776 lines for the previous version against 730 lines for this revised version that is a cut of 6% in length). In the following responses we referred to the marked version of the revised MS.

**REFEREE#1**

**Comment#1:**

Major Comments

I appreciate authors' efforts in addressing the previous issues.

**Answer#1:**

We thank the Referee for his/her overall positive feedbacks concerning our MS.

**Comment#2:**

I am sorry but I still have some concerns that need to be addressed. First, I still think the Result section is way too long. Authors need to carefully edit the section to make it easy for readers to follow, while keeping information to length ratio in mind.

**Answer#2:**

We agree with the referee's comment, also shared by the other referee. First, we moved the parts L.252-269 (water mass balance explanation), 366-397 (carbon export explanation), 421-429 (degassing explanation) from the Results section to the Materials and Methods section L.340-358 (water mass balance explanation), L.360-391 (we created in the Materials and Methods a section 2.6 entitled carbon stocks in groundwater, exports to streams and degassing to the atmosphere). Second, we considerably reduced the use of abbreviations in the text. Only DIC and DOC abbreviations are still used and thus we deleted the table 1 that contained a list of abbreviations used in our MS. Third, we deleted and/or shortening some useless parts of the Results section (L.405-409, 421-432, 464-465, 488-490, 501-505, 508-511; 513-515) and the Discussion section (L.876-881). We also made the whole MS easier to follow, revising the style and making the discussion much more focused on the new finding of our study

**Comment#3:**

Second, authors have divided the study period into several seasons/flow conditions but do they really have enough sample size to understand the catchment processes for each season or flow condition? In my opinion, this is a limitation of the study and should be clearly mentioned in the paper.

**Answer#3:**

We agree with the referee's comment that sampling frequency is one limitation of the study because we have few carbon samples in each hydrological season (Tab. S1). Indeed, carbon sampling less time-spaced (e.g., weekly or bimonthly) could have improved the quantification of carbon exports accounting for sudden hydrological events. However, our sampling frequency allowed a relatively precise description and understanding of the mechanisms involved in carbon mobilization in soils and groundwaters. The observed trends were consistent from one hydrological cycle to another and from one sampling station to another the second year of monitoring (Fig. 4, 5). Because of the characteristics of the soil and watershed, changes in DIC/DOC concentrations in groundwater and streams are relatively slow. Moreover, as we observed in the figure 5 of the MS, temporal evolution of DIC/DOC in groundwater and streams is quite well described. In the revised MS, we specified those

important points L.465-468, 746-756 and we also added in the revised table 4 (page 66) the number (N) of samples for each hydrological period and for each sampled piezometer and stream.

**Comment#4:**

Third, there are still lots of typos that need to be fixed. Specific Comments: Introduction-Line 49 Authors could add some more references as this was established long ago.

**Answer#4:**

We revised the sentence as follows (L.63-66):

Indeed, biogeochemical cycling within and across the terrestrial–aquatic interface is dynamically linked to the water cycle (Johnson et al., 2006; Battin et al., 2009), because dissolved carbon is primarily mobilized and transported by the movement of water (Hope et al., 1994; Hagerdon et al., 2000; Kawasaki et al., 2005).

**Comment#5:**

Line 76- Would "provide" be a better choice? Enable doesn't seem appropriate here?

**Answer#5:**

We replaced the word "enable" with "provide". L.108

**Comment#6:**

Line 85- Odd word usage- "migration of water table"?

**Answer#6:**

This sentence was deleted in the revised MS (L.119)

**Comment#7:**

Line 84-90- The language of the objectives could be tightened as they are long and very descriptive. I would recommend turning these loose objectives into more concrete and impactful questions.

**Answer#7:**

We agree with the referee comment and thus re-wrote the objectives as follows: "In this study, we instrumented a temperate watershed that offers the convenience of a homogeneous lithology (permeable sandy soil), vegetation (pine forest) and topography (very flat coastal plain), as well as a simple hydrological functioning exclusively as shallow groundwater drainage. This simple configuration with no surface runoff allows us to identify what are the main factors that control the DIC/DOC leaching to streams, the DIC:DOC ratio in groundwater and streams, and their variation in space and over time. At the plot scale, we relate DIC and DOC temporal dynamics in groundwater with hydrology and metabolic activity of the forest ecosystem. At the watershed scale, we quantify DIC and DOC transfers through the groundwater-stream interface, and we describe the fate of this carbon in first-order streams"

See changes at L.114-130

**Comment#8:**

Line 113- Can we use " recharge" instead of fuels?

**Answer#8:**

We modified as suggested. L.165

**Comment#9:**

Line 246- drainage "enrichment"? What does it mean?

**Answer#9:**

The drainage enrichment parameter was defined as the ratio between two stream drainages of successive orders. However, we re-wrote this paragraph in a more specific way and deleted this confusing term (L.319-337).

**Comment#10:**

Line 249- "Low/lower" order not "inferior" order.

**Answer#10:**

We-rewrote this part in a more specific way (L.319-337).

**Comment#10:**

Line 250 It would be nice to add few text on how authors got "2.3 fold". Citing an unpublished paper doesn't help here and several other places in the draft.

**Answer#10:**

We found this comment a bit confusing because we already written an entirely paragraph to explain our methodology to estimate groundwater inputs in first-order stream at the scale of the watershed. We re-wrote this part in a more specific way (L.319-337) and we hope that it is now clear enough. Note that our companion paper containing a full table with all drainage calculations has been accepted with minor revision in Journal of Hydrology a couple of weeks ago, so we hope to be able to provide a doi in the published version of the present paper.

**Comment#11:**

Results- It is confusing and distracting to have results and methods together in the same section. Both together have made results way longer than what it should have been without methods.

**Answer#11:**

We agree with the referee's comment and thus we moved the parts L.252-269 (water mass balance explanation), 366-397 (carbon export explanation), 421-429 (degassing explanation) from the Results section to the Materials and Methods section L.340-358 (water mass balance explanation), L.360-391 (we created in the Materials and Methods a section 2.6 entitled carbon stocks in groundwater, exports to streams and degassing to the atmosphere).

**Comment#12:**

Line 275- 305 Too much text has been dedicated to a single figure. Most of the information shared in the text can be easily read and understood from the figure 2.

**Answer#12:**

We agree with the referee's comment and we shortened the section 3.1 (Hydrological parameters and water mass balance). See changes L.391-443.

**Comment#13:**

It is unclear that why author choose to show precipitation on a negative scale? Is it because of the input versus output?

**Answer#13:**

As it suggested by the referee we revised this figure, precipitation is on a positive scale, drainage and evapotranspiration are on a negative scale, groundwater storage could be on a negative or a positive scale depending if the groundwater gained or lose water during the month. See changes at page 73.

**Comment#14:**

Discussion-Line 438- I am not sure this statement is entirely true. The study was conducted for 18 months but there are several gaps in the collected datasets. As per Table S1, there are only 6 occasions when the datasets were available for all groundwater wells and stream. Is it

OK to divide such short period (18 months) in multiple flow conditions and seasons when some of the flow conditions have 1-2 GW observations?

**Answer#14:**

As we explained in the MS in the 2.5 (Hydrological monitoring) section (L.301) the 4 hydrological parameters (P, D, ETR and GWS) used in the hydrological mass balance originated from continuous measurements with a daily timescale for D, and with a half hour timescale for ETR, P and GWS. We chose to present the water mass balance in a monthly timescale (Fig. 2) to be consistent with our monthly timescale carbon sampling. There is no gap in our hydrological dataset. Moreover, the objective of the present study was not to describe very precisely the water budget at short timescale but rather to understand and quantify the water budget at monthly timescales to discuss DIC/DOC variations at the same monthly timescale. We made some changes L.610-613 in order to satisfy this comment.

**Comment#15:**

Line 454- Please edit this sentence - "transpiration flow through plants and the evapotranspiration were maximum…."?

**Answer#15:**

We edited this sentence as follows: "The evapotranspiration was high during growing season and late summer periods when the precipitations were low" (L.631-631)

**Comment#16:**

Line 496- Please edit this sentence! It may make sense to start the sentence with " During base flow conditions…"

**Answer#16:**

First, note that we rewrote more clearly the 4.2 section. We edited this sentence as follows: "During base flow conditions, the DOC concentrations in groundwater were relatively stable at our study sites, even after rainy periods (Tab. 4; Figs. 2c, 5c), which suggests that soil DOC

in upper horizons was not preferentially mobilized to groundwater by rainwater infiltration".
See changes at L.772-775.

**Comment#17:**

Line 490-Line 533 It would be nice to break this 1.5 page long paragraph into more readable
2-3 paragraphs

**Answer#17:**

We agree with this comment and we did the same for the rest of the text (see as an example
the 2.1 and 4.2 sections at page 7 and 38).

**Comment#18:**

Figures-Fig1 – Please add units to topography in the map legend.

**Answer#18:**

The figure 1 has been edited. We added units to topography in the map legend (page 72)

**Comment#19:**

Fig7 – Please Fix typo in GS column- decreasing

**Answer#19:**

We fixed typo as it was suggested (Page 78)

**REFEREE#2**

**Comment#1:**

General comment

The authors well considered and responded to the comments by reviewers. I think this version of manuscript is acceptable after the minor revision about the following points.

**Answer#1:**

We thank the Referee for his/her overall positive feedbacks concerning our MS.

**Comment#2:**

However, to tell the truth, I hope to the authors more clearly address the message what we, the readers of BG, should learn from this study in conclusion. The conclusion of this version only shows the results of the Leyre catchment. Is this just a case study? Please consider this point again before preparing the final version of manuscript

**Answer#2:**

We agree with the referee's comment when he/she mentions that our conclusion was too much focused on the studied Leyre catchment. However, the description of mechanisms that mobilize and export dissolved C from the soils will be useful for all the community working on soil carbon. In addition, our quantitative results can be easily compared to other lowland environments having shallow groundwater as well as other podzols ecosystems which has a surface of 485 million ha (3% of the land surface area) worldwide (Driessen et al., 2000).

We re-wrote part of the discussion and the conclusion insisting on the processes occurring in the soil and groundwater and how these processes compare with the literature in similar or different environments. See changes at L.842-871, 963-977, 989-1021.

**Comment#3:**

Reduce the abbreviations. For example, TA is less used. HF, GS, LS, and EW (L.275-) is never used in other sections, e.g. L.351.

**Answer#3:**

We agree with the referee's comment and thus we reduced the abbreviations in the MS. In the revised MS, only DIC and DOC abbreviations are still used in the text and thus we deleted the table 1 that listed the abbreviations of the MS.

**Comment#4:**

L.366-397 and L.421-430. These parts should be move to method section.

**Answer#4:**

We agree with the referee's comment, also shared by the other referee. First, we moved the parts L.252-269 (water mass balance explanation), 366-397 (carbon export explanation), 421-429 (degassing explanation) from the Results section to the Materials and Methods section L.340-358 (water mass balance explanation), L.360-391 (we created in the Materials and Methods a section 2.6 entitled carbon stocks in groundwater, exports to streams and degassing to the atmosphere).

**Comment#5:**

L.617. Remove 'with'

**Answer#5:**

We removed this term as suggested. See changes at L.950

[revised manuscript text omitted]

Table S2: Periods of calculation for C stocks, C exports and C degassing. X corresponds to a calculation. [a, b, c] for these periods the day of sampling of surface waters do not correspond exactly to the day of sampling of groundwater (Tab. S1). C stocks in groundwater can be calculated only for Bilos plot since we do not have data about the total height of the permeable layer in the other plots. HF, GS, LS and EW represent high flow, growing season, late summer and early winter periods, respectively.